# FREQUENCY-AWARE DYNAMIC GAUSSIAN SPLATTING

**Qiaowei Miao**,* **Jinsheng Quan**,* **Kehan Li, Yichao Xu, Yi Yang, Yawei Luo**†
Zhejiang University

## ABSTRACT

We present **Frequency-Aware Dynamic Gaussian Splatting (FAGS)**, a novel approach to mitigating motion blur in 4D reconstruction, particularly under novel viewpoints. This blur stems from a fundamental spectral conflict in existing methods, which struggle to **balance high-frequency rendering details with high-frequency motion.** FAGS addresses this challenge with two key innovations. First, we introduce a frequency-differentiated Gaussian kernel that refines the alpha-blending process of 3D Gaussian Splatting. By adaptively classifying Gaussians into two types—a slowly varying kernel for smooth, low-frequency regions and a sharp-transitioning kernel for high-frequency boundaries—our method explicitly separates representation responsibilities, preserving fine details without sacrificing continuity. Second, we propose a Fourier-Deformation Network that enhances motion expressiveness. This network employs high-frequency Fourier embeddings to capture diverse motion patterns by learning amplitudes across frequency components. To further improve accuracy, we integrate a frequency-aware gate in fusion module, which predicts and regulates the relative deformation of each Gaussian. Extensive experiments on both synthetic and real-world 4D benchmarks demonstrate that FAGS significantly reduces motion blur and enhances structural details, achieving state-of-the-art performance.

## 1 INTRODUCTION

Recent advances in deformation-field-driven **3D Gaussian Splatting (3DGS)** have achieved impressive results in 4D reconstruction Yang et al. (2024); Wu et al. (2024a); Zhou et al. (2024); Huang et al. (2024); Lin et al. (2024); Li et al. (2024); Xie et al. (2024); Kratimenos et al. (2023); You & Hou (2024); Lu et al. (2024); Luiten et al. (2023); Guo et al. (2024). Despite this progress, dynamic reconstructions often suffer from pronounced **motion blur**, especially under novel viewpoints. These existing approaches attempt to alleviate this issue by refining deformation fields to better align dynamic content across frames. However, a **core limitation lies in balancing high-frequency rendering details and high-frequency motion**.

We analyze the balancing problem between Gaussians and Deformation Network. As illustrated in Fig. 1(a), vanilla 3DGS adopts a fixed opacity profile for each Gaussian—opaque at the center and gradually transparent toward the boundary. While this design works well for static scenes by relying on overlapping Gaussians, it severely limits expressiveness in dynamic settings. The fixed profile forces the deformation network to handle two contradictory tasks simultaneously: on one hand, it must orchestrate dense Gaussian stacking to recover high-frequency appearance at individual time steps; on the other hand, it must drive these Gaussians coherently across multiple frames to model high-frequency motion without causing scattering. This dual burden often biases the network toward uniform low-frequency motion as a compromise, which ultimately results in severe motion blur under novel views. As shown in Fig. 1(b), a more effective solution is to enhance Gaussian expressiveness for high-frequency details and to empower the deformation network with frequency-aware flexibility, thereby separating the responsibilities of representation and motion modeling.

In this work, we tackle this problem from a **frequency perspective**, introducing **Frequency-Aware Dynamic Gaussian Splatting (FAGS)** model. Our approach explicitly decouples high-frequency detail representation from high-frequency motion modeling through two key components. First, the **Frequency-Differentiated Gaussian Kernel** employs a learnable alpha modulation function that adaptively adjusts each Gaussian's projection. This allows Gaussians to specialize as either (a) sharp high-frequency kernels for detail boundaries or (b) smooth low-frequency kernels for uniform regions,

---

*These authors contributed equally to this work.
†Corresponding author.

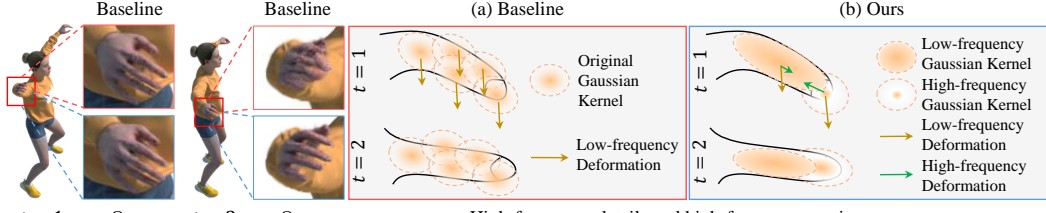

Figure 1: Analysis of high-frequency details and motion. (a) Baseline: Standard Gaussian kernels require dense overlapping to capture high-frequency details, which complicates deformation control and biases the network toward smooth, low-frequency trajectories, resulting in motion blur. (b) Our method: Gaussians in high- and low-frequency regions are differentiated, enabling the deformation network to capture high-frequency motion on top of global low-frequency deformations, allowing each Gaussian to follow its own fine-grained dynamics.

thereby reducing reliance on dense Gaussian stacking. Second, the **Fourier-Deformation Network** models motion trajectories as a superposition of periodic movements at multiple frequencies. By learning the amplitude distribution of high- and low-frequency components, it enables precise motion estimation—capturing rapid, high-frequency displacements over short intervals while maintaining stable, low-frequency trends globally. Our main contributions are as follows:

- We identify motion blur in 4D reconstruction as arising from a fundamental spectral conflict between balancing high-frequency rendering details and high-frequency motion.

- We propose the **Frequency-Differentiated Gaussian Kernel**, which uses adaptive alpha modulation to allow Gaussians to specialize as high- or low-frequency kernels, separating the modeling of high-frequency details from smooth low-frequency regions.

- We design the **Fourier-Deformation Network**, incorporating high-frequency Fourier features and a frequency-aware gate to simultaneously capture high-frequency local deformations while suppressing unnecessary motion in low-frequency points.

- Extensive experiments on synthetic and real-world 4D benchmarks show that FAGS effectively reduces motion blur and sharpens structural details.

## 2 RELATED WORKS

**Dynamic Neural Radiance Fields.** Neural Radiance Fields (NeRF) have emerged as a powerful paradigm for representing scenes as continuous volumetric functions. Early extensions to dynamic settings aimed to reconstruct 4D scenes. Neural 3D Video Li et al. (2022), for example, introduces temporal latent variables combined with hierarchical training and importance sampling, achieving scalable performance and establishing DyNeRF as a benchmark for dynamic NeRF. Another prominent line of work leverages deformation fields to extend NeRF into the temporal domain, as in Nerfies Park et al. (2021a) and D-NeRF Pumarola et al. (2021). Nerfies decomposes 4D space into a static NeRF template with time-dependent deformation fields, while D-NeRF directly predicts spatial displacements over time. Li et al. (2021) further encode both geometry and motion by training a network to learn scene flow, thereby capturing spatial structure alongside temporal dynamics.

**Dynamic Gaussian Splatting.** Recent research extends static 3D Gaussian Splatting (3DGS) into the 4D domain Yang et al. (2024); Wu et al. (2024a); Zhou et al. (2024); Huang et al. (2024); Lin et al. (2024); Miao et al. (2025b; 2024; 2025a); Quan et al. (2025), enabling efficient modeling of object motion and scene dynamics. By combining anisotropic Gaussians with deformation fields, these methods achieve high-quality rendering with real-time efficiency Wu et al. (2024b). Existing approaches fall into two categories: *iterative* and *deformation-based*. Iterative methods update Gaussian parameters frame by frame, such as D-3DGS Luiten et al. (2023) which propagates Gaussians across frames while enforcing temporal coherence. While conceptually simple, they often struggle with occlusions and unseen regions, and usually require multi-camera setups. In contrast, deformation-based methods adopt a shared canonical representation and use deformation fields to predict temporal offsets, avoiding per-frame optimization and improving scalability. For example, Yang et al. (2024) learn spatial and temporal adjustments via an MLP queried by 4D coordinates, while Wu et al. (2024a) integrate a spatio-temporal encoder–decoder for efficient, high-quality

modeling. These approaches are generally more scalable and robust, though they depend heavily on accurate deformation modeling.

Despite these advances, prior work mainly focuses on refining deformation fields or canonical representations while overlooking the *spectral characteristics of motion*. In particular, balancing high-frequency motion with high-frequency image detail remains unresolved, often leading to motion blur in novel views. Our work addresses this gap by introducing a frequency-aware formulation that explicitly disentangles these spectral demands within the Gaussian representation.

## 3 METHODOLOGY

### 3.1 PRELIMINARIES

**3D Gaussian Splatting.** 3D Gaussian Splatting Kerbl et al. (2023) (3DGS) represents a scene as a set of anisotropic 3D Gaussian primitives parameterized by position $\mu \in \mathbb{R}^3$, rotation $R$, scaling $S$, spherical harmonic coefficients $c_{SH}$ for view-dependent color, and opacity $o$. The covariance is defined as $\Sigma^{3D} = RSS^T R^T$ to ensure positive semi-definiteness. Given a camera view, each Gaussian is projected to the image plane with center $\mu^{2D}$ and covariance $\Sigma^{2D}$, and the final pixel color is rendered via alpha blending:

$$C(\mathbf{p}) = \sum_{i=1}^{N} c_i \alpha_i \prod_{j=1}^{i-1} (1-\alpha_j), \quad \alpha_i = o_i \exp\left[ -\tfrac{1}{2}(p - \mu^{2D}i)^T (\Sigma^{2D}i)^{-1}(p - \mu_i^{2D})\right], \quad (1)$$

where $N$ denotes the number of Gaussians overlapping pixel $p$. This formulation enables high-quality, differentiable rendering suitable for optimization.

**Deformation Network.** Deformation networks Wu et al. (2024a); Xu et al. (2024) model temporal evolution of Gaussians in dynamic scenes. Specifically, given a Gaussian's position $x$ and timestamp $t$, an MLP predicts residual updates $(\Delta_x, \Delta_R, \Delta_S)$ to its position, rotation, and scaling. The deformed Gaussian $(x + \Delta_x, R + \Delta_R, S + \Delta_S, c_{SH}, o)$ is then rasterized using Eq. 1. This framework maintains a shared canonical representation and improves temporal coherence compared to per-frame optimization. However, since deformation primarily modifies geometry while alpha blending handles color composition, it struggles to represent high-frequency motion and sharp appearance changes, often leading to motion blur, especially near object boundaries or during rapid deformations.

### 3.2 FREQUENCY-DIFFERENTIATED GAUSSIAN KERNEL

The limited expressiveness of individual Gaussians often forces the deformation network to rely on overlapping kernels, biasing deformation fields toward real motion. To address this, we propose the **Frequency-Differentiated Gaussian Kernel** (FDGK), which endows each Gaussian with an adaptive alpha modulation function. This enables specialization into high-frequency Gaussians for complex textures and low-frequency Gaussians for smooth regions, reducing overlap and stabilizing deformation optimization. We realize this extension of Gaussian splatting through three straightforward modifications, detailed below.

Recalling Eq. 1, $\alpha$ decay is primarily governed by the subsequent power operation $g$, making it sensitive to numerical fluctuations. We generalize the computation as

$$\alpha_i = \min(o_i \psi(g), 0.99), \quad \psi(g) = rg + b, \quad \text{where } g = \exp\left[ -\frac{1}{2}(\mathbf{p} - \boldsymbol{\mu}_i^{2D})^T (\boldsymbol{\Sigma}_i^{2D})^{-1}(\mathbf{p} - \boldsymbol{\mu}_i^{2D})\right]. \tag{2}$$

Here, the slope $r$ modulates $\alpha$'s smoothness ($r \to 1$) versus sharpness ($r \to \infty$), effectively generating high- and low-frequency Gaussians, with a standard Gaussian corresponding to $r = 1$ and $b = 0$. While DRK Huang et al. (2025) also adjusts $r$ to enhance expressiveness, the mapping boundaries are implicitly determined by $r$, so Gaussians sharing the same slope inherit identical differentiated regions. This coupling constrains the separation between high- and low-frequency Gaussians. In contrast, FDGK introduces a learnable boundary parameter $\beta$ for each Gaussian, dynamically controlling its differentiation region independently of $r$, thereby enabling each Gaussian to adapt its frequency characteristics and differentiation span and substantially improving representational flexibility. We provide more detailed results to illustrate the effects of $\beta$ in Appn. G.

First, to simplify the analysis, we partition the original mapping range into three intervals and constrain the central interval to pass through (0.5, 0.5). This construction enables us to focus on the middle region for Gaussian differentiation. Building on this foundation, we introduce the adaptive alpha modulation function $\psi(g)$:

$$
\psi(g) = \begin{cases} \frac{0.5+\lambda-0.5\beta-\lambda\beta}{0.5+\lambda-\beta} g, & g \in [0, p_l), \\ (0.5+\lambda)g + (0.25-0.5\lambda), & g \in [p_l, p_r], \\ \frac{0.5+\lambda-0.5\beta-\lambda\beta}{0.5+\lambda-\beta}(g-1)+1, & g \in (p_r, 1], \end{cases} \qquad \begin{cases} p_l = \max(\frac{0.5+\lambda-\beta}{1+2\lambda}, 0), \\ p_r = \min(\frac{0.5+\lambda+\beta}{1+2\lambda}, 1). \end{cases} \tag{3}
$$

Here, $p_l$ and $p_r$ denote the left and right endpoints of the central mapping interval, defining the middle range, as illustrated in Fig. 2. We introduce a controlling parameter $\lambda \in [0, 1]$ to adjust the slope of the mapping within this interval. Specifically, the slope and offset are defined as $r = 0.5 + \lambda$ and $b = 0.25 - 0.5\lambda$, with $\lambda = 0.5$ serving as the demarcation to differentiate high- and low-frequency Gaussians. Fig. 2 visualizes how varying $\lambda$ affects 3D Gaussians: when $\lambda = 0.5$, the Gaussians correspond to standard 3D Gaussians with a gradual transparency gradient from the projection center outward; for $\lambda < 0.5$,

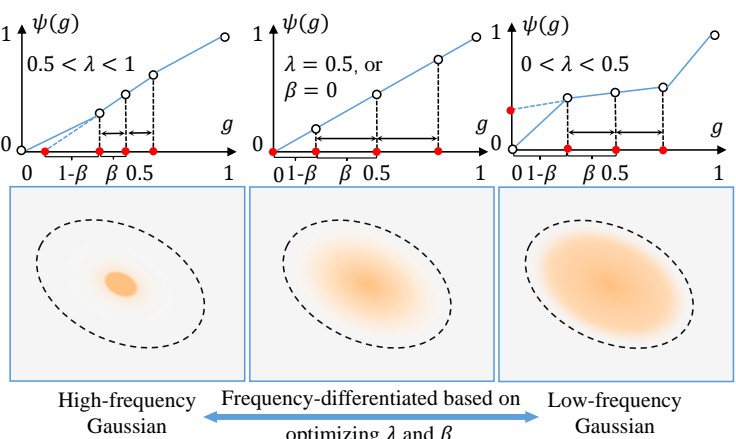

High-frequency Gaussian — Frequency-differentiated based on optimizing $\lambda$ and $\beta$ — Low-frequency Gaussian

Figure 2: Adaptive alpha modulation function of the frequency-differentiated Gaussian kernel. The piecewise activation function $\psi(g)$ regulates the $\alpha$ values of Gaussian projections, with its slope modulated by the learnable parameters $\lambda$ and $\beta$. By optimizing these parameters, the activation becomes differentiated, enabling Gaussians to specialize into low-frequency and high-frequency representations.

$\psi(g)$ varies slowly with $g$, producing smooth, uniform transparency typical of low-frequency Gaussians; for $\lambda > 0.5$, $\psi(g)$ changes rapidly, yielding sharp transitions and distinct boundaries characteristic of high-frequency Gaussians.

To further control the **differentiated region**, we introduce a boundary parameter $\beta$. Let $g_0$ denote the point where $\psi(g_0) = 0$. The distance $d_{g_0}$ between $g_0$ and 0.5 defines the allowable span of the frequency-differentiated region. We scale this distance by $\beta$ to determine the left and right endpoints as $p_l = 0.5 - \beta d_{g_0}$ and $p_r = 0.5 + \beta d_{g_0}$, as formulated in Eq. 3. Optimizing $\lambda$ and $\beta$ allows each Gaussian to adaptively regulate its frequency and span.

The values of $\lambda$ and $\beta$ for each Gaussian can be directly applied in the rasterization process and gradient backpropagation. As a continuous and differentiable piecewise function spanning the full range of $g \in [0, 1]$, $\psi(g)$ preserves the opacity representation of Gaussians. The $\lambda$ and $\beta$ can be optimized by backpropagation:

$$
\frac{\partial \psi(g)}{\partial \lambda} = \begin{cases} \frac{\beta^2-\beta}{(0.5+\lambda-\beta)^2}g, & g \in [0, p_l) \\ g - 0.5, & g \in [p_l, p_r] \\ \frac{\beta^2-\beta}{(0.5+\lambda-\beta)^2}(g-1), & g \in (p_r, 1] \end{cases} \qquad \frac{\partial \psi(g)}{\partial \beta} = \begin{cases} \frac{0.25-\lambda^2}{(0.5+\lambda-\beta)^2}g, & g \in [0, p_l) \\ 0, & g \in [p_l, p_r] \\ \frac{0.25-\lambda^2}{(0.5+\lambda-\beta)^2}(g-1), & g \in (p_r, 1] \end{cases} \tag{4}
$$

By jointly optimizing $\lambda$ and the boundary parameter $\beta$, the activation function $\psi(g)$ is adaptively learned, differentiating Gaussians into high- and low-frequency types while stabilizing $\alpha$ values within a controlled range, thereby ensuring robust and stable training of the deformation network.

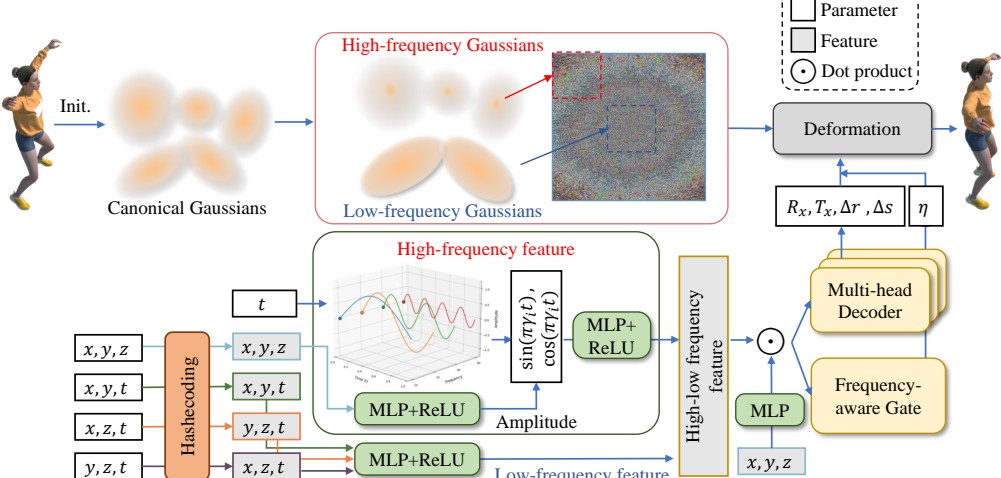

Figure 3: **Overview of Frequency-Aware Gaussian Splatting. Top**: Gaussians are initialized in a canonical state and progressively differentiate into low- and high-frequency types during optimization, fitting low-frequency smooth regions and high-frequency details, respectively. **Bottom**: Coordinates $x, y, z$ and time $t$ are hash-encoded to generate low-frequency features, which are then fused with high-frequency Fourier embeddings to form a unified high–low frequency representation. This representation is processed by a multi-head decoder to predict deformations, with a frequency-aware gate modulating motion strength. Finally, rendering Gaussians to produce the final outputs.

### 3.3 FOURIER-DEFORMATION NETWORK

Existing deformation network Xu et al. (2024); Yang et al. (2024); Wu et al. (2024a); Cao & Johnson (2023) typically apply hash encoding to 4D coordinates $(x, y, z, t)$ and use MLP to model the motion of Gaussian points. While hash encoding preserves strong spatiotemporal continuity and maintains geometric consistency in the reconstruction, these methods remain limited in modeling rapid temporal motion. This inherent trade-off between capturing high-frequency spatial details and high-frequency motion often leads to motion blur.

To address this, we propose the Fourier-Deformation Network, which explicitly models high-frequency temporal variations for each Gaussian. Directly optimizing trajectories over all time steps is computationally prohibitive for scenes with over 100k Gaussians. Instead, we encode high-frequency motion for each point, integrate it with low-frequency spatial features, and leverage the resulting high–low frequency feature to efficiently predict Gaussian deformations.

Specifically, we decompose the 4D coordinate $(x, y, z, t)$ into four 3D hash encodings: $(x, y, z)$, $(x, y, t)$, $(y, z, t)$, and $(x, z, t)$, capturing different spatial-temporal interactions. These embeddings produce a spatial feature $f_{\text{spa}}$ and temporal feature $f_{\text{tem}}$ via MLP:

$$f_{\text{spa}} = \text{MLP}(H_{xyz}(x, y, z)), \quad f_{\text{tem}} = \text{MLP}\big(\text{concat}\big[H_{xyt}(x, y, t), H_{xzt}(y, z, t), H_{yzt}(x, z, t)\big]\big). \tag{5}$$

**High-frequency Fourier Embedding.** We formulate each Gaussian's motion as a segment of an infinitely repeating cycle, allowing its trajectory to be decomposed through Fourier analysis into amplitude–frequency components. To better capture high-frequency temporal variations, we design a dedicated high-frequency Fourier embedding for each Gaussian:

$$f_{\text{fre}} = [w_1 \sin(\pi\gamma_1 t), \quad w_1 \cos(\pi\gamma_1 t), \quad \ldots, \quad w_m \sin(\pi\gamma_m t), \quad w_m \cos(\pi\gamma_m t)]^{\text{T}}, \tag{6}$$

where $\gamma_i = 2^{\frac{3i-3}{m-1}}, m = \frac{D}{2}$. $D$ denotes the dimension of $f_{\text{spa}}$. This temporal encoding adopts denser sampling designed to effectively span multiple temporal scales. The amplitude vector $[w_1, \ldots, w_m]$ is specific to each Gaussian and time-invariant, predicted from its spatial feature $f_{\text{spa}}$ via an MLP:

$$[w_1, \ldots, w_m] = \text{MLP}(f_{\text{spa}}). \tag{7}$$

The resulting Fourier feature $f_{\text{fre}}$ captures periodic high-frequency motion, which can be aligned or scaled to complement low-frequency temporal embeddings $f_{\text{tem}}$. By fusing $f_{\text{fre}}$ with low-frequency features, the deformation network achieves precise control over both slow and rapid motions, enabling accurate and stable reconstruction of dynamic scenes.

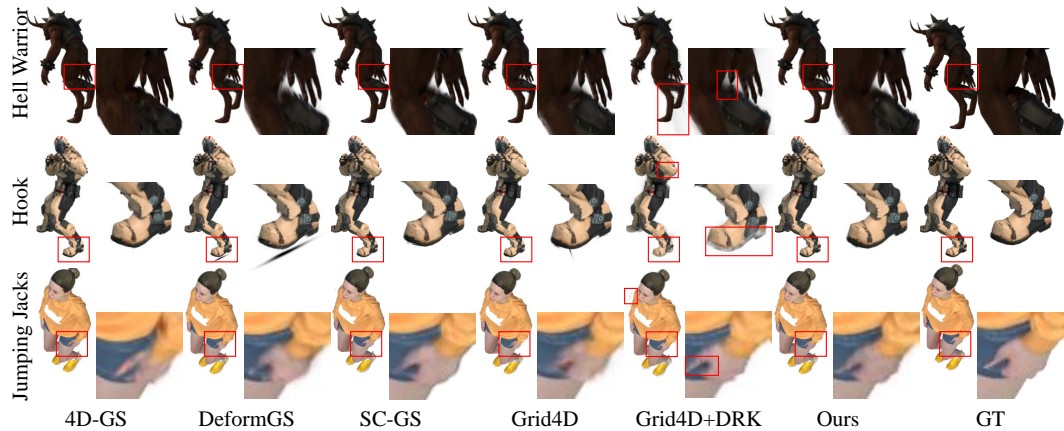

Figure 4: Qualitative results on the synthetic D-Nerf dataset.

**Frequency-aware Gate.** A deformation network inherently predicts updates for all Gaussians, which may inadvertently modify points that should remain static. To selectively suppress low-frequency motion while promoting genuine high-frequency deformations, we introduce a frequency-aware gate that estimates each Gaussian's relative motion state at each time step, *i.e.,* whether the point is more dynamic or more static compared to other Gaussians. The high-frequency Fourier feature $f_{\text{fre}}$ is fused with the temporal embedding $f_{\text{tem}}$ (*i.e.,* the low-frequency features), and the resulting representation is used to predict the deformation parameters via the deformation network $D_\theta$:

$$\mu' = \eta R_x \mu + \eta T_x, \quad S' = S + \eta \Delta s, \quad R' = R + \eta \Delta r, \quad D_\theta(f_{\text{FD}}) = \{\eta, R_x, T_x, \Delta r, \Delta s\}, \quad (8)$$

where $R_x$ and $T_x$ denote the predicted rotation and translation, and $\eta$ is the frequency-aware gate score that modulates the deformation strength at time $t$. By adaptively weighting the updates, the gate assigns larger $\eta$ values to Gaussians undergoing high-frequency motion, allowing rapid changes in their attributes, while near-static, low-frequency Gaussians receive smaller $\eta$ values, effectively suppressing undesired motion and stabilizing the optimization process.

## 3.4 OPTIMIZATION

**Fourier Frequency Loss**. Although the frequency-differentiated Gaussian kernels and the Fourier-Deformation Network are designed to capture high-frequency details and motion, their potential cannot be fully realized without a frequency-aware optimization objective. To this end, we introduce a Fourier frequency loss. Specifically, for a rendered image $I'$ and its target image $I$, we use Fast Fourier Transform (FFT) to obtain the corresponding amplitude spectra $I'_{\text{amp}}$ and $I_{\text{amp}}$. The Fourier frequency loss is then defined as:

$$\mathcal{L}_{\text{fre}} = \|I'_{\text{amp}} - I_{\text{amp}}\|_1. \quad (9)$$

Since the phase primarily encodes structural information and the rendered image $I'$ shares highly similar geometry with the target image $I$, the differences in phase are minimal; therefore, we only consider discrepancies in amplitude. We integrate $\mathcal{L}_{\text{fre}}$ with the reconstruction losses used in Grid4D Xu et al. (2024), yielding the final training objective:

$$\mathcal{L} = \sigma_c \mathcal{L}_{L1} + (1 - \sigma_c)\mathcal{L}_{\text{MISS}} + \sigma_r \mathcal{L}_r + \sigma_{\text{fre}}\mathcal{L}_{\text{fre}}, \quad (10)$$

where $\sigma_c$, $\sigma_r$, and $\sigma_{\text{fre}}$ are hyperparameters balancing the contributions of each term. Here, $\mathcal{L}_r$ denotes the smooth regularization loss used in Grid4D. Following Xu et al. (2024), we keep $\sigma_c$ and $\sigma_r$ consistent with their settings and fix them during training. Similar to prior works Yang et al. (2024); Wu et al. (2024a), we initialize with static canonical Gaussians at the beginning of training to ensure stable convergence. For Gaussians initialized from SfM Schonberger & Frahm (2016), we shorten the static initialization stage for improved efficiency. Moreover, we adopt the adaptive density controller and opacity resetting mechanism introduced in Gaussian Splatting Kerbl et al. (2023). The overall pipeline of our FAGS is illustrated in Fig. 3.

Table 1: Quantitative results on D-NeRF dataset. Best and second-best results are highlighted.

| Model | Bouncing Balls PSNR↑ | SSIM↑ | LPIPS↓ | Hell Warrior PSNR↑ | SSIM↑ | LPIPS↓ | Hook PSNR↑ | SSIM↑ | LPIPS↓ | Jumping Jacks PSNR↑ | SSIM↑ | LPIPS↓ |
|---|---|---|---|---|---|---|---|---|---|---|---|---|
| Tensor-4D* | 24.47 | 0.962 | 0.044 | 31.26 | 0.925 | 0.074 | 28.63 | 0.943 | 0.064 | 24.20 | 0.925 | 0.067 |
| HexPlane* | 40.36 | 0.992 | 0.031 | 24.30 | 0.944 | 0.073 | 28.26 | 0.955 | 0.052 | 31.74 | 0.974 | 0.036 |
| TiNeuVox-B | 40.28 | 0.991 | 0.039 | 28.30 | 0.966 | 0.064 | 31.86 | 0.972 | 0.044 | 34.77 | 0.987 | 0.033 |
| DeformGS | 41.67 | 0.995 | 0.008 | 41.09 | 0.986 | 0.027 | 37.06 | 0.986 | 0.016 | 37.06 | 0.986 | 0.016 |
| 4D-GS | 40.77 | 0.994 | 0.016 | 35.37 | 0.985 | 0.020 | 32.90 | 0.977 | 0.027 | 35.37 | 0.985 | 0.020 |
| SC-GS | 41.85 | 0.996 | 0.022 | 42.10 | 0.989 | 0.028 | 38.81 | 0.991 | 0.014 | 39.80 | 0.992 | 0.013 |
| Grid4D | 42.36 | 0.995 | 0.007 | 42.85 | 0.990 | 0.016 | 38.89 | 0.990 | 0.009 | 39.36 | 0.993 | 0.008 |
| Grid4D+DRK | 40.82 | 0.995 | 0.011 | 41.92 | 0.989 | 0.019 | 36.07 | 0.983 | 0.021 | 37.57 | 0.991 | 0.011 |
| Ours | 43.32 | 0.996 | 0.007 | 43.13 | 0.991 | 0.014 | 40.18 | 0.992 | 0.008 | 40.52 | 0.994 | 0.006 |

| Model | Mutant PSNR↑ | SSIM↑ | LPIPS↓ | Standup PSNR↑ | SSIM↑ | LPIPS↓ | Trex PSNR↑ | SSIM↑ | LPIPS↓ | Average PSNR↑ | SSIM↑ | LPIPS↓ |
|---|---|---|---|---|---|---|---|---|---|---|---|---|
| Tensor-4D* | 29.11 | 0.945 | 0.060 | 30.56 | 0.958 | 0.036 | 31.25 | 0.964 | 0.048 | 28.50 | 0.946 | 0.056 |
| HexPlane* | 33.66 | 0.982 | 0.028 | 34.12 | 0.983 | 0.019 | 31.01 | 0.976 | 0.028 | 31.92 | 0.972 | 0.038 |
| TiNeuVox-B | 33.99 | 0.977 | 0.030 | 36.10 | 0.986 | 0.020 | 32.87 | 0.978 | 0.031 | 34.02 | 0.979 | 0.037 |
| DeformGS | 42.13 | 0.994 | 0.007 | 43.70 | 0.994 | 0.009 | 37.78 | 0.993 | 0.010 | 40.07 | 0.991 | 0.013 |
| 4D-GS | 37.71 | 0.988 | 0.016 | 37.88 | 0.990 | 0.014 | 34.14 | 0.985 | 0.022 | 36.30 | 0.986 | 0.019 |
| SC-GS | 42.55 | 0.995 | 0.011 | 46.16 | 0.997 | 0.008 | 39.53 | 0.994 | 0.010 | 41.59 | 0.994 | 0.015 |
| Grid4D | 43.94 | 0.996 | 0.004 | 46.51 | 0.997 | 0.004 | 40.01 | 0.994 | 0.008 | 41.99 | 0.993 | 0.008 |
| Grid4D+DRK | 38.95 | 0.988 | 0.019 | 42.64 | 0.992 | 0.012 | 38.04 | 0.993 | 0.012 | 39.43 | 0.990 | 0.015 |
| Ours | 44.16 | 0.997 | 0.004 | 47.30 | 0.997 | 0.003 | 40.69 | 0.995 | 0.008 | 42.76 | 0.995 | 0.007 |

Table 2: Quantitative results on Neu3D dataset. The color marks the best and the second best.

| Model | Coffee Martini PSNR↑ | SSIM↑ | LPIPS↓ | Cook Spinach PSNR↑ | SSIM↑ | LPIPS↓ | Cut Roasted Beef PSNR↑ | SSIM↑ | LPIPS↓ |
|---|---|---|---|---|---|---|---|---|---|
| 4D-GS | 27.00 | 0.885 | 0.189 | 31.80 | 0.942 | 0.159 | 32.06 | 0.939 | 0.167 |
| Grid4D | 27.90 | 0.893 | 0.179 | 32.38 | 0.947 | 0.144 | 33.20 | 0.947 | 0.157 |
| Ours | 28.37 | 0.899 | 0.179 | 32.79 | 0.967 | 0.145 | 33.20 | 0.950 | 0.148 |

| Model | Flame Steak PSNR↑ | SSIM↑ | LPIPS↓ | Sear Steak PSNR↑ | SSIM↑ | LPIPS↓ | Average PSNR↑ | SSIM↑ | LPIPS↓ |
|---|---|---|---|---|---|---|---|---|---|
| 4D-GS | 30.23 | 0.945 | 0.147 | 32.18 | 0.958 | 0.144 | 30.65 | 0.934 | 0.161 |
| Grid4D | 32.07 | 0.942 | 0.136 | 32.61 | 0.955 | 0.132 | 31.63 | 0.937 | 0.149 |
| Ours | 32.95 | 0.947 | 0.132 | 33.58 | 0.968 | 0.125 | 32.18 | 0.946 | 0.146 |

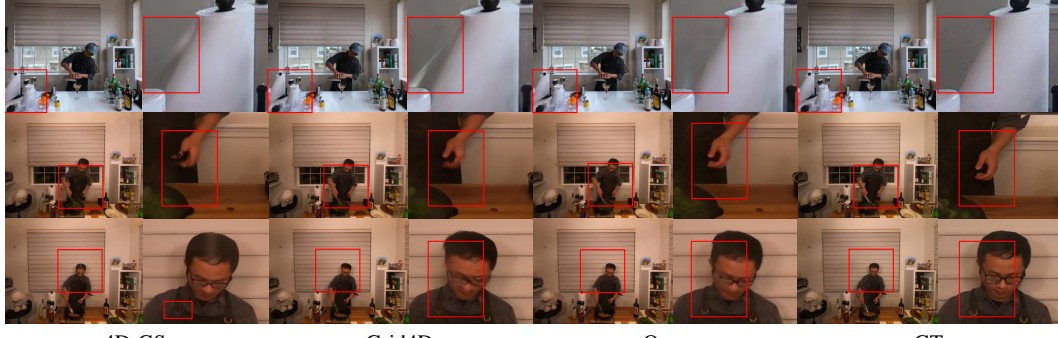

| 4D-GS | Grid4D | Ours | GT |
|---|---|---|---|

Figure 5: Qualitative results on the real-world Neu3D dataset.

## 4 EXPERIMENTS

**Datasets.** We evaluate FAGS on D-NeRF Pumarola et al. (2021), HyperNeRF Park et al. (2021b), and Neu3D Li et al. (2022). D-NeRF is a synthetic monocular dataset with accurate time-varying poses, while HyperNeRF and Neu3D contain real-world captures with poses estimated via COLMAP Fisher et al. (2021). Rendering resolutions are $800 \times 800$, $536 \times 960$, and $1,352 \times 1,024$, respectively.

**Baselines.** We compare against state-of-the-art 4D reconstruction methods Wu et al. (2024a); Yang et al. (2024); Huang et al. (2024); Xu et al. (2024); Fang et al. (2022); Shao et al. (2023); Cao & Johnson (2023). NeRF-based approaches include Tensor-4D Shao et al. (2023) and TiNeuVox Fang et al. (2022) (plane- and grid-based explicit representations). Gaussian-based models include 4D-GS Wu et al. (2024a), DeformGS Yang et al. (2024), and SC-GS Huang et al. (2024), with the latter extending DeformGS via sparse control points. Grid4D Xu et al. (2024) improves deformation control using a 4D decomposed hash encoding with attention.

**Implementation Details.** Experiments are performed on an NVIDIA 3090 GPU. We train using Adam with a learning rate of $0.002$ for $\lambda$ and $\beta$. FAGS is implemented in CUDA/C++ and integrates with Grid4D Xu et al. (2024) as the deformation backbone. We set $\sigma_{\text{fre}} = 0.3$ as the default hyperparameter. Other settings that we follow Xu et al. (2024)'s work.

Table 3: Quantitative results on real-world HyperNeRF dataset, including 4 rig subsets (Rig) and the 6 interpolation subsets (Interpolation) . The color marks the best and the second best .

| Model | Interpolation (6 scenes) | | | | Rig (4 scenes) | | | |
|---|---|---|---|---|---|---|---|---|
| | PSNR ↑ | SSIM ↑ | LPIPS ↓ | MS-SSIM↑ | PSNR ↑ | SSIM ↑ | LPIPS ↓ | MS-SSIM↑ |
| TiNeuVox | 27.11 | - | - | 0.923 | 24.21 | - | - | 0.836 |
| 4D-GS | 27.58 | 0.794 | 0.315 | 0.910 | 24.64 | 0.665 | 0.363 | 0.654 |
| Grid4D | 28.59 | 0.844 | 0.199 | 0.993 | 25.24 | 0.685 | 0.319 | 0.818 |
| Ours | 29.02 | 0.850 | 0.195 | 0.994 | 25.63 | 0.719 | 0.269 | 0.885 |

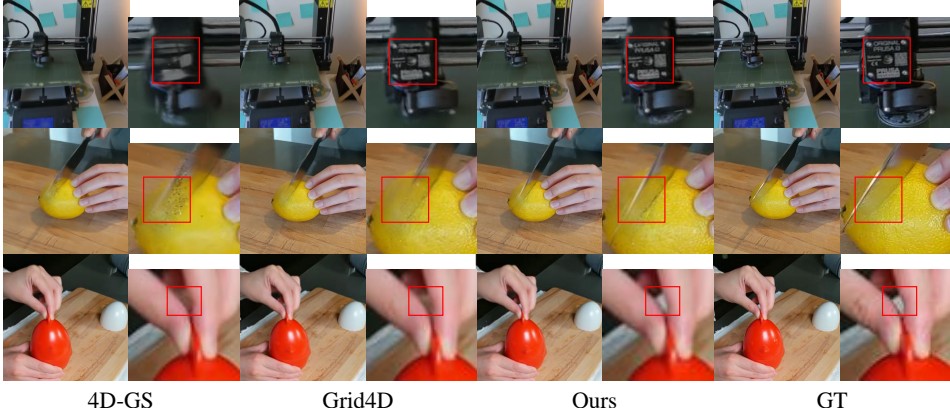

|       4D-GS       |       Grid4D       |       Ours       |       GT       |

Figure 6: Qualitative results on the real-world HyperNeRF dataset.

## 4.1 EXPERIMENTAL RESULTS

We evaluated FAGS against state-of-the-art methods on the synthetic D-NeRF dataset and real-world HyperNeRF and Neu3D datasets. Quantitative metrics include PSNR Fardo et al. (2016), SSIM Wang et al. (2003), LPIPS Zhang et al. (2018), and MS-SSIM Wang et al. (2003). Our Frequency-Differentiated Gaussian Kernel and Fourier Deformation Network improve rendering of fine textures and achieve significant quantitative gains (Tab. 1, Tab. 2, Tab. 3). Across datasets, our method consistently improves visual fidelity. On D-NeRF (Fig. 4), object boundaries are better preserved, *e.g.,* feet in Hook and hand details in JumpingJacks. In Neu3D (Fig. 5), baselines with non-differentiated Gaussians often misadjust displacements, causing blurred static backgrounds and loss of high-frequency motion details. In HyperNeRF (Fig. 6), improper handling of multiple Gaussians leads to artifacts such as disappearing knife tips. By contrast, our FAGS effectively address these issues, demonstrating superior control over both spatial structure and motion details.

## 4.2 ABLATION STUDIES

We present the ablation results of each component in Tab. 4 and Fig. 8. **(a) Frequency-Differentiated Gaussian Kernels (FDGK).** By decoupling Gaussians into high- and low-frequency categories, FDGK reduces redundant stacking and alleviates the deformation network's burden. This specialization enhances the reconstruction of fine-grained high-frequency details, as evidenced by sharper hand and foot structures in Fig. 8. Compared to standard 3DGS, FDGK provides greater flexibility and efficiency in representing frequency-aware scene components. **(b) High-Frequency Fourier Embedding (HFE).** HFE injects temporally varying high-frequency signals into each Gaussian, preventing uniformly smooth low-frequency motion. This enables better separation and preservation of fine details, particularly in articulated regions such as fingers and leg joints. **(c) Frequency-aware Gate (FG).** FG adaptively regulates the deformation strength of each Gaussian based on its frequency-aware features. Low-frequency Gaussians are suppressed to remain stable, while genuine high-frequency motions are amplified, improving the network's control over dynamic regions. Unlike hard clamping methods that enforce static bounds, we let $\eta$ smoothly vary, allowing the deformation network to adaptively modulate each Gaussian's dynamic behavior at every timestep. **(d) Fourier Frequency Loss ($\mathcal{L}_{\mathrm{fre}}$).** This loss explicitly regularizes the spectral domain by emphasizing hard-to-optimize high-frequency regions. It guides both Gaussians and the deformation network toward precise local fitting, yielding reconstructions closer to ground truth with enhanced detail fidelity.

Table 4: Quantitative ablation results on the synthetic D-NeRF dataset.

| Model | Full | w/o FDGK | w/o HFE | w/o FG | w/o $\mathcal{L}_{\text{fre}}$ | w/o (FG + $\mathcal{L}_{\text{fre}}$) | w/o FDGK.$\lambda$ | w/o FDGK.$\beta$ |
|---|---|---|---|---|---|---|---|---|
| PSNR ↑ | 42.76 | 42.11 | 42.38 | 42.70 | 42.50 | 42.43 | 42.30 | 42.26 |
| SSIM ↑ | 0.995 | 0.993 | 0.994 | 0.994 | 0.994 | 0.994 | 0.994 | 0.994 |
| LPIPS ↓ | 0.007 | 0.009 | 0.009 | 0.008 | 0.008 | 0.008 | 0.008 | 0.008 |

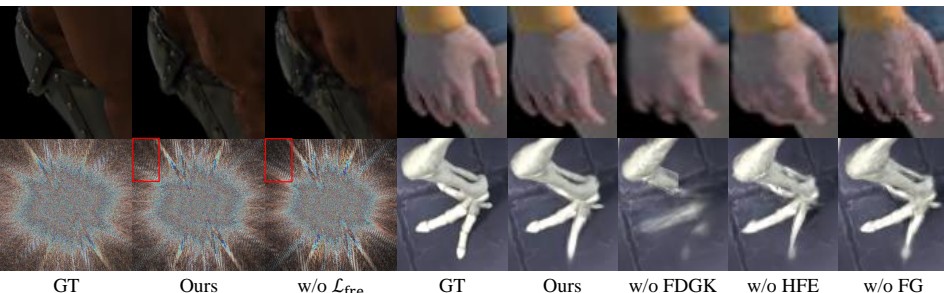

| GT | Ours | w/o $\mathcal{L}_{\text{fre}}$ | GT | Ours | w/o FDGK | w/o HFE | w/o FG |

Figure 8: Ablation study visualization results on D-Nerf dataset.

## 4.3 MORE DISCUSSION

**High–Low Frequency Gaussians ($\lambda$'s distribution).** At initialization, all Gaussian kernels are assigned $\lambda = 0.5, \beta = 0.5$, which is equivalent to the standard 3D Gaussian kernel. After about 5,000 training steps, the Gaussians clearly bifurcate into two distinct groups corresponding to low- and high-frequency components, with a ratio of roughly 3:2, as shown in Fig. 7(a). More importantly, once this differentiation emerges, the two groups remain stable throughout subsequent training, indicating that the frequency-differentiated Gaussian kernel can both converge rapidly and maintain consistent separation. For more details, please refer to the Appn. G.

**Boundary ($\beta$'s distribution).** The introduced boundary parameter $\beta$ exhibits significant distributional shifts in the early stages of training, as shown in Fig. 7(b). It suggests that the model actively leverages $\beta$ to rapidly adapt the differentiated region for better fitting. This highlights the crucial role of $\beta$ in enriching the expressiveness of frequency-differentiated Gaussian kernel.

**Frequency-Based Motion Differentiation (Gate's distribution).** During motion learning, we observe that Gaussians are not simply classified into either completely static or fully dynamic states, as shown in Fig. 7(c). Instead, guided by the Fourier-Deformation Network, each Gaussian is adaptively assigned a relative motion state according to its frequency characteristics. A large proportion of Gaussians concentrate in the low-frequency regime, while a

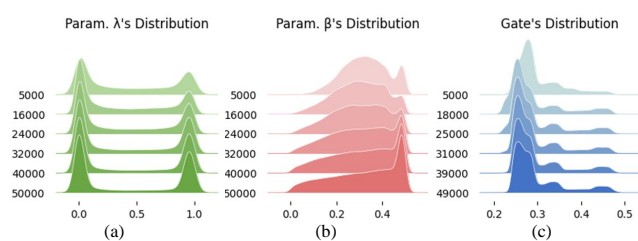

Figure 7: Distributions of frequency-differentiated parameters $\lambda$, $\beta$, and the frequency-aware gate across training steps.

smaller subset exhibits high-frequency behavior, demonstrating the network's ability to capture motion at multiple scales and achieve frequency-based differentiation of Gaussian dynamics.

**FDGK vs. DRK.** To thoroughly evaluate reconstruction performance, we integrate the DRK Huang et al. (2025) with the Grid4D Xu et al. (2024) deformation network and test on the D-NeRF dataset. Models are trained using the default DRK parameters, and results are shown in Fig. 1. Compared to the original Grid4D, the more flexible DRK kernels do not consistently improve reconstruction quality and often degrade it in several scenes. We attribute this to the excessive flexibility of DRK, which makes it difficult for the deformation network to control Gaussian deformations across multiple time steps. In contrast, FDGK undergoes clear kernel differentiation early in training and exhibits stable behavior throughout the optimization process (Fig. 7), which in turn enables more reliable control of temporal deformations.

**Reconstruction under Motion Blur.** We evaluate FAGS under severe motion blur using the real-world motion blur dataset from BARD-GS Lu et al. (2025), and directly compare its performance with BARD-GS. In this dataset, the images are synthetically blurred for reconstruction, while evaluation

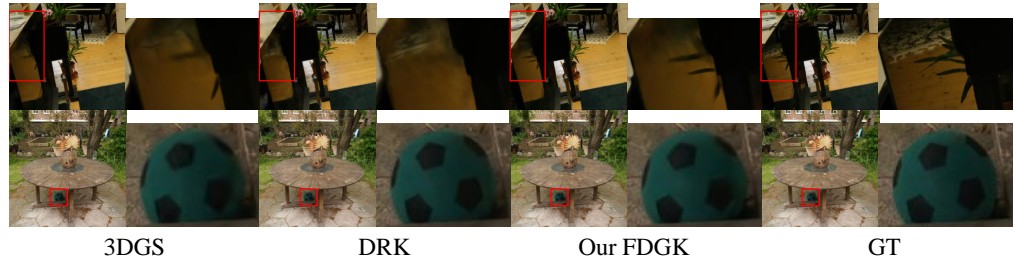

| 3DGS | DRK | Our FDGK | GT |

Figure 9: Qualitative results on the Mip360-NeRF 3D dataset.

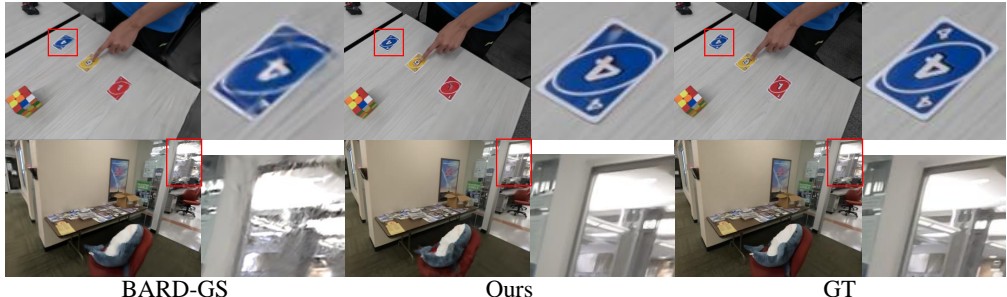

| BARD-GS | Ours | GT |

Figure 10: Qualitative comparisons on the motion-blur dataset.

is performed against the sharp ground-truth frames. Quantitative results are reported in Tab. 7, and qualitative comparisons are shown in Fig. 10. Despite the challenging blur and without deblurring pre-processing with MPRNet Mehri et al. (2021)), FAGS remains competitive, often producing substantially sharper and more faithful reconstructions. Applying a pre-processing deblurring step with MPRNet can further improve the results, yielding performance surpassing BARD-GS.

**Quantifying High-frequency Motion.** In Fig. 11, we show the temporal power spectrum of Gaussian' trajectories in the "Jumpingjacks" sequence. The temporal power spectrum shows that FAGS concentrates substantially more Gaussians in the high-frequency band than Grid4D.

This indicates that the FDN actively induces rapid deformations on many Gaussians, effectively capturing high-frequency motion. Furthermore, we provide heatmaps of the temporal second-order finite-difference energy in Fig. 15. Compared to Grid4D, FAGS exhibits stronger contrast between bright and dark regions, indicating that the FDN accurately captures the differ-

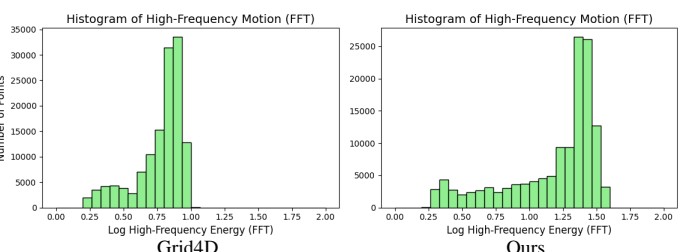

Figure 11: Temporal Motion Power Spectrum of Gaussians.

ences in deformations across consecutive time steps, effectively avoiding overly smooth, global motion. These quantitative results demonstrate that FAGS can precisely model and preserve high-frequency motions, which are essential for accurate dynamic reconstruction.

## 5 CONCLUSION

In this paper, we present **Frequency-Aware Dynamic Gaussian Splatting (FAGS)**, a principled approach to mitigating motion blur in 4D reconstruction. The blur arises from a spectral conflict in conventional methods that couple high-frequency rendering with high-frequency motion, leading to optimization instability. FAGS resolves this challenge with two complementary innovations. First, a **Frequency-Differentiated Gaussian Kernel** refines alpha blending by explicitly decoupling opacity from Gaussian shape, allowing Gaussians to specialize in either smooth, low-frequency regions or sharp, high-frequency boundaries. Second, a **Fourier-Deformation Network** enhances motion representation by leveraging high-frequency Fourier embeddings and a frequency-aware gate, enabling accurate and expressive motion modeling. Extensive experiments on both real and synthetic 4D benchmarks show that FAGS effectively reduces motion blur, sharpens structural details.

## 6 ACKNOWLEDGMENT

This work was supported by National Natural Science Foundation of China (62293554, U2336212), National Science and Technology Major Project (2023ZD0121300), "Pioneer" and "Leading Goose" R&D Program of Zhejiang (2025C02022), Zhejiang Provincial Natural Science Foundation of China (LZ24F020002), Zhejiang Provincial Natural Science Foundation of China under Grant No. LDT23F02023F02, Ningbo Innovation "Yongjiang 2035" Key Research and Development Programme (2024Z292), and Young Elite Scientists Sponsorship Program by CAST (2023QNRC001). The author gratefully acknowledges the support of Zhejiang University Education Foundation Qizhen Scholar Foundation.

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
