# - Supplementary -

For a better understanding of the main paper, we provide additional details in this supplementary material, which is organized as follows:

- §F shows the deblur results on BARD-GS real world dataset.

## A STRUCTURE DETAILS OF FOURIER DEFORMATION NETWORK

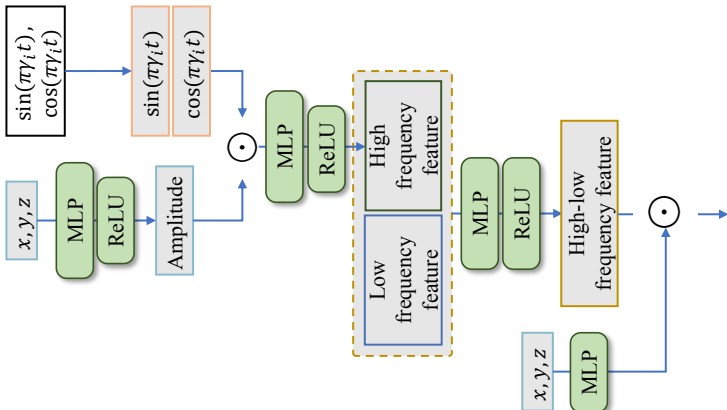

Figure 12: Structural Details of High- and Low-Frequency Feature Extraction and Fusion.

In Fig. 12, we provide a detailed illustration of the high–low frequency acquisition and fusion mechanism. Based on the spatial features $f_{\mathrm{spa}}$ (*i.e.,* $x, y, z$ in Fig. 12), we first predict the amplitudes. For the sine and cosine encodings with the same index $i$, the amplitude is shared. By concatenating these encodings and passing them through an MLP followed by a ReLU activation, the high-frequency features are projected to the same dimensionality as the low-frequency features. The two are then concatenated and processed by another MLP with dimensionality reduction and ReLU activation, yielding the final high–low frequency feature representation. This representation is fused with spatial features via an attention mechanism to form the spatiotemporal features.

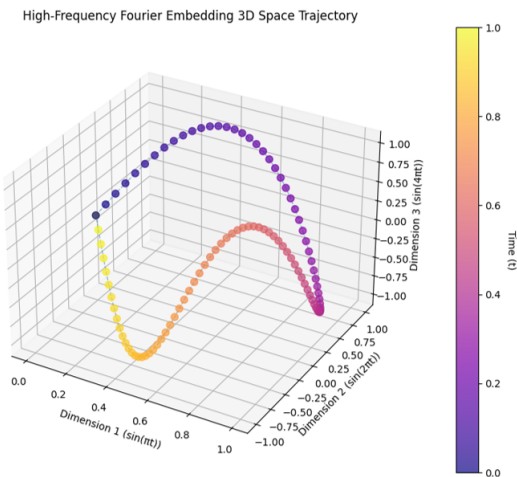

Figure 13: Low-Dimensional High-Frequency Fourier Embedding in 3D Space.

# B HIGH-FREQUENCY FOURIER EMBEDDING

To provide a more intuitive illustration of the temporal encoding capability of the High-Frequency Fourier Embedding under different periodic functions, we reduce its dimensionality to three and map the components onto the $(x, y, z)$ coordinates, visualizing the trajectory of time from $t = 0$ to $t = 1$. Specifically, we set $x = \sin(\pi t)$, $y = \sin(2\pi t)$, and $z = \sin(4\pi t)$. The resulting trajectory shows that each time step corresponds to a unique point that encodes frequency information, while the representation remains continuous over time. In Eq. 6, the actual embedding has a higher dimensionality and additionally incorporates cosine terms, providing denser periodic cues that help the deformation network capture high-frequency details. In Fig. 13, we visualize the trajectory of the high-frequency Fourier embedding in a low-dimensional 3D space. Despite combining multiple high-frequency periodic functions, the embedding does not exhibit overlaps within a single period, effectively preventing multi-mapping of temporal coordinates.

# C QUANTITATIVE ANALYSIS OF MOTION FREQUENCY

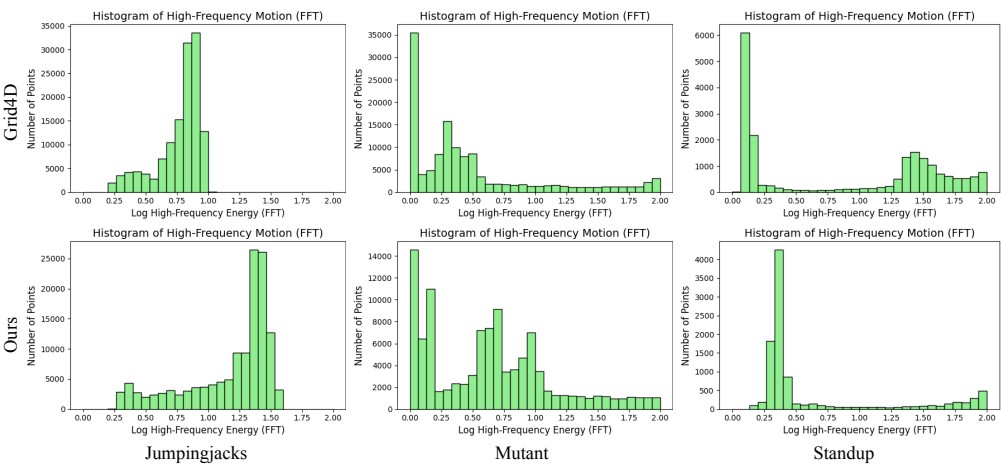

Figure 14: Distributions of high-frequency motion energy between Grid4D and our FAGS.

To further demonstrate the effectiveness of FAGS in capturing high-frequency motion, we present the high-frequency motion energy distribution and the comparative heatmaps of temporal second-order finite-difference energy.

**High-Frequency Motion Energy Distribution.** We record the 3D coordinates of each Gaussian at every time step across the full temporal span and apply a Fourier transform to their motion trajectories to obtain the frequency-domain energy spectrum. We then take the logarithm of the energy for enhanced visualization. As shown in Fig. 14, the horizontal axis represents frequencies from low to high, while the vertical axis corresponds to the number of Gaussians. Compared with Grid4D, FAGS exhibits a higher overall motion frequency distribution and contains noticeably more Gaussians with relatively high-frequency motion components.

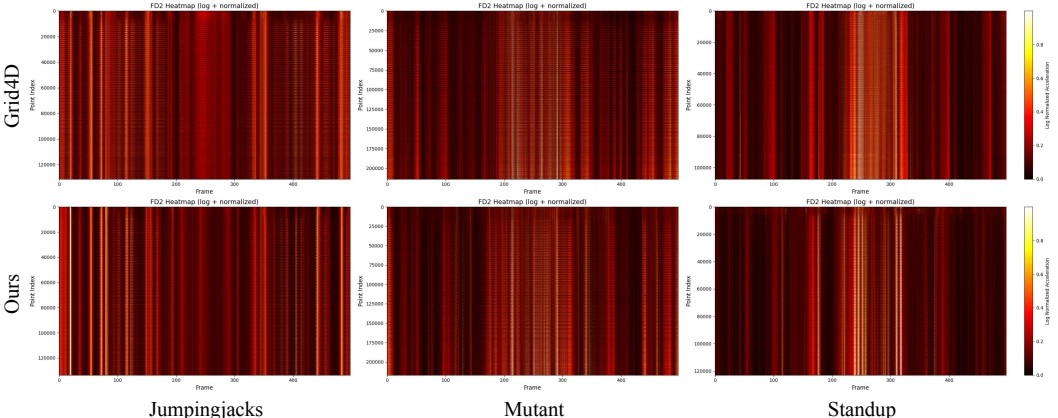

Figure 15: Heatmaps of temporal second-order finite-difference energy for Grid4D and our FAGS.

**Temporal Second-order Finite-difference Energy.** In Fig. 15, the horizontal axis represents time steps and the vertical axis corresponds to Gaussian indices. The color intensity reflects the magnitude of the temporal second-order finite-difference energy. Compared with Grid4D, FAGS exhibits stronger contrast patterns at the same time steps, suggesting that it captures motion variations between adjacent control Gaussians more effectively. This implies that FAGS provides finer and more precise control over Gaussian deformation, rather than relying on overly smooth, globally uniform motion.

## D  ANALYSIS OF SPECTRAL CONFLICT

Table 5: Conflict risk quantitative results on D-Nerf dataset.

| Model | Bouncing Balls | | Hell Warrior | | Hook | | Jumping Jacks | |
|---|---|---|---|---|---|---|---|---|
| | PSNR ↑ | Conflict Risk↓ | PSNR ↑ | Conflict Risk↓ | PSNR ↑ | Conflict Risk↓ | PSNR ↑ | Conflict Risk↓ |
| Grid4D | 42.36 | 15.3% | 42.85 | 69.1% | 38.89 | 42.0% | 39.36 | 36.9% |
| Ours | 43.32 | 9.8% | 43.13 | 36.2% | 40.18 | 33.2% | 40.52 | 11.9% |
| **Model** | Mutant | | Standup | | Trex | | Average | |
| | PSNR ↑ | Conflict Risk↓ | PSNR ↑ | Conflict Risk↓ | PSNR ↑ | Conflict Risk↓ | PSNR ↑ | Conflict Risk↓ |
| Grid4D | 43.94 | 44.9 % | 46.51 | 41.0% | 40.01 | 35.2% | 41.99 | 40.6% |
| Ours | 44.16 | 29.7% | 47.30 | 36.2% | 40.69 | 24.0% | 42.76 | 25.8% |

Spectral conflict arises from the competition between Gaussians and the deformation network in jointly encoding color. As illustrated in Fig. 2, the deformation network manipulates the Gaussians to represent color, while gradients are backpropagated to optimize both components simultaneously. This can lead to conflicting updates, where Gaussians and the deformation network compete to encode high-frequency details.

To measure this effect, we introduce a gradient-based metric called Conflict Risk. For each Gaussian, we compute gradients of the training loss obtained from rendered images on the D-NeRF test set. If a Gaussian's gradient exceeds the densify and prune threshold, it is considered at risk of conflict. The Conflict Risk is then defined as the proportion of such Gaussians. Intuitively, a large gradient indicates that the Gaussian cannot be fully controlled by the deformation network to render high-quality novel views and may require cloning or splitting, even though it fits the training views accurately.

We report Conflict Risk and PSNR in Tab. 5. The lower Conflict Risk correlates with higher PSNR, suggesting that this metric effectively captures the detrimental interaction between Gaussians and the deformation network and provides a useful indicator of high-frequency motion fidelity.

## E  FDGK VS. DRK

Table 6: Quantitative evaluation on Mip-NeRF360 dataset Barron et al. (2022).

| Model | PSNR↑ | LPIPS↓ | SSIM ↑ |
|---|---|---|---|
| 3DGS | 28.20 | 0.24 | 0.93 |
| DRK | 28.15 | 0.24 | 0.92 |
| Our FDGK | 28.32 | 0.26 | 0.92 |

**4D scenes.** In this work, we focus on improving reconstruction quality in 4D scenes. For a broader comparison, we combine the DRK Huang et al. (2025) Gaussians with the Grid4D deformation network. The quantitative results are reported in Tab. 1. For a more intuitive comparison, visualizations are provided in Fig. 4.

**3D scenes.** Fig. 9 shows the visual comparison between FDGK and DRK in 3D scenes, and the corresponding quantitative results are presented in Tab. 6. We trained and evaluated DRK using its official open-source implementation. Compared with DRK, FDGK demonstrates improvements in 3D scene reconstruction.

By comparing FDGK and DRK in both 4D and 3D reconstruction scenarios, we observe that FDGK consistently enhances reconstruction quality. The improvement is particularly pronounced in 4D scenes when combined with our FDN module.

## F  DEBLUR RESULTS ON BARD-GS REAL WORLD DATASET

Table 7: Quantitative evaluation on BARD-GS Real World Dataset. $*$ indicates reported results, while # denotes reproduced results.

| Model | PSNR↑ | SSIM ↑ | LPIPS↓ | Time ↓ |
|---|---|---|---|---|
| DyBluRF* | 20.57 | 0.761 | 0.361 | - |
| DeformGS* | 22.83 | 0.824 | 0.350 | - |
| DeformGS + MPRNet* | 23.10 | 0.836 | 0.263 | - |
| 4DGS* | 22.06 | 0.809 | 0.373 | - |
| 4DGS + MPRNet* | 22.58 | 0.829 | 0.286 | - |
| BARD-GS* | 25.13 | 0.851 | 0.156 | - |
| BARD-GS# | 23.46 | 0.847 | 0.183 | 380m |
| Ours | 23.12 | 0.839 | 0.201 | 45m |
| Ours + MPRNet | 25.42 | 0.873 | 0.135 | 50m |

In Tab. 7, we compare our method with BARD-GS as well as other baselines reported in the literature. Our approach outperforms all baselines except BARD-GS and achieves results comparable to it,

demonstrating the ability to handle severely motion-blurred inputs without relying on a dedicated deblurring pre-processing model. Notably, as illustrated in Fig. 10, our method often preserves finer details than BARD-GS. In addition, it offers a clear advantage in computational efficiency.

## G   FREQUENCY-DIFFERENTIATED GAUSSIAN DISTRIBUTION

To better illustrate the differentiation of Gaussians and the deformation network's capability in capturing motion variations, Fig. 16 shows the distributions of the frequency-differentiated parameters $\lambda$ and $\beta$, along with the frequency-aware gate, on the D-NeRF dataset. For $\lambda$, we clearly observe that in all scenes, Gaussians are effectively separated into distinct frequency bands, with the distributions exhibiting pronounced peaks at both ends. The introduction of the boundary parameter $\beta$ further enhances flexibility by adaptively controlling the range of frequency differentiation. As illustrated in the second row, Gaussians in different scenes learn diverse boundary parameters, leading to significant distributional variations. This indicates that $\beta$, in conjunction with $\lambda$, enables more fine-grained scene fitting. In addition, the frequency-aware gate shows that the deformation network does not enforce a rigid separation of all Gaussians into static (0) or fully dynamic (1). Instead, it adaptively identifies the relative motion state of each Gaussian point within the scene, which is a more reasonable modeling choice. Notably, in the Trex scene, the moving dinosaur and the static table surface form two distinct peaks, corresponding to low-frequency and high-frequency motion. This highlights the capability of our High-frequency Fourier Network to more accurately capture relative motion.

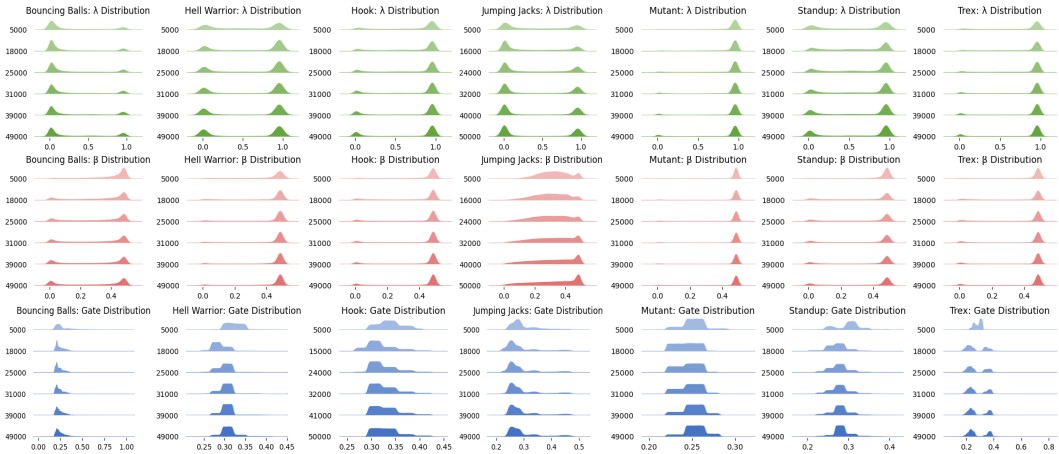

Figure 16: The distribution of $\lambda$, $\beta$ and frequency-aware gate in D-Nerf dataset.

## H   BALANCING HIGH-FREQUENCY DETAILS AND HIGH-FREQUENCY DYNAMICS

To visually illustrate the trade-off in previous methods between capturing high-frequency details and high-frequency dynamics, we track fine high-frequency Gaussian points on the hand in the *jumpingjacks* scene, as shown in Fig. 1. As depicted in Fig. 17, during training, the deformation network tends to densely stack Gaussians to fit local high-frequency details. At novel time steps and viewpoints, these high-frequency points, initially concentrated on the hand, are driven to drift toward the arm position in order to fit the arm's high-frequency motion. In contrast, our method effectively maintains the consistency of the motion trajectories. Furthermore, in the more rigid calf region, the Gaussians tracked with the Frequency-Differentiated Gaussian Kernel (FDGK) show much more consistent aggregation between their initial and final positions compared to the case without frequency differentiation. These two examples indicate that combining the deformation network with a frequency-differentiated Gaussian kernel enables the network to learn more accurate and discriminative motion information. By precisely capturing high-frequency motion for each Gaussian, the deformation network preserves trajectory consistency. This observation directly informed the

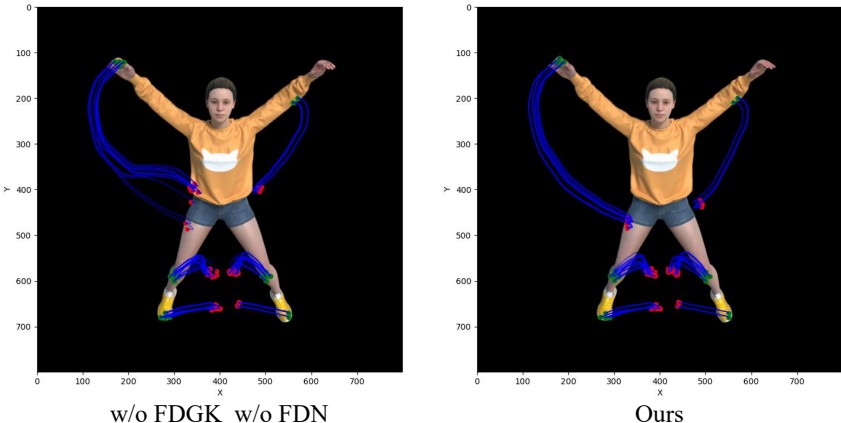

w/o FDGK  w/o FDN                    Ours

Figure 17: Trajectory visualization of Gaussians. Without frequency-differentiated Gaussian kernel, the deformation network drives some Gaussian points to drift, attempting to fit other regions.

design of both our Gaussian kernel and the deformation network, and demonstrates that our method better balances high-frequency detail with high-frequency dynamics.

# I  MORE VISUALIZATION RESULTS

To further illustrate the regional representations captured by FAGS, we visualize the high-low

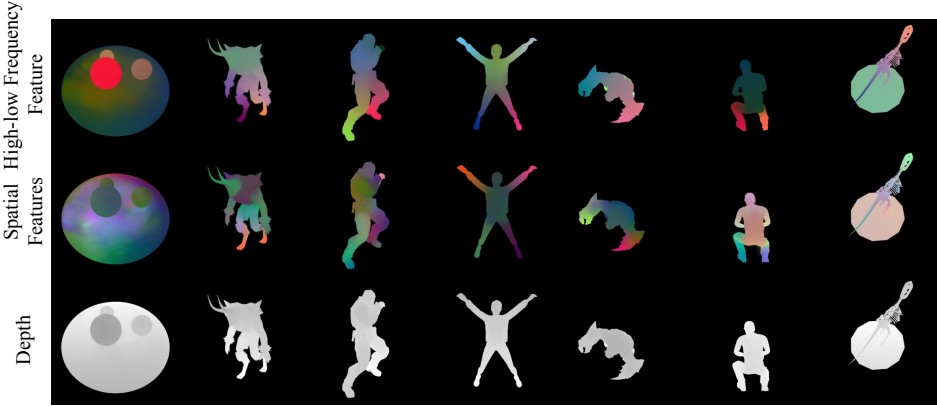

Figure 18: High-low frequency features, spatial features, and depth visualization.

frequency features, spatial features, and depth maps in Fig. 18. In the high- and low-frequency features (first row), regions with the same color correspond to similar low-frequency motion, while variations in hue within the same color indicate high-frequency motion states of the Gaussians. The visualization demonstrates that our method effectively distinguishes motion states across different regions and refines high-frequency motion details. The second row shows the visualization of each Gaussian's static attributes, where local regions exhibit consistent spatial representations. This aligns with our design, indicating that the deformation network preserves spatial structure rather than being disrupted by high-frequency fitting. The third row presents depth information, where our method produces compact density distributions and smooth yet detail-preserving surfaces.

# J  LOCAL DEPTH VISUALIZATION

Relying on the stacking of high- and low-frequency Gaussian points to fit colors significantly

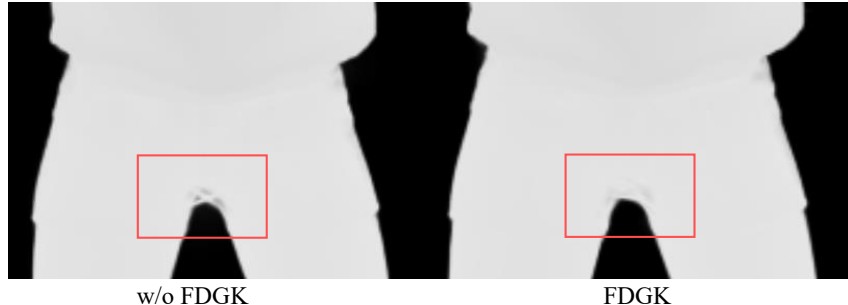

w/o FDGK        FDGK

Figure 19: When frequency-differentiated Gaussian kernel is absent, depth map observations distinctly reveal insufficient local compactness.

increases the training difficulty for the deformation network. This not only leads to drifting but also results in areas with poor compactness. While the difference in the jeans area is barely perceptible to the eye in Fig. 17, observing with a depth map clearly reveals the discrepancies in compactness. As shown in Fig. 19, without frequency-differentiated Gaussian kernel, the deformation network struggles to stack Gaussian points uniformly. In contrast, when frequency-differentiated Gaussian kernel is incorporated, these Gaussian points are differentiated and remain stable during later training stages, reducing their reliance on stacking to fit the region. This, in turn, yields a smoother and more uniform rendering result on the depth map.

## K    More Additional Ablations

Table 8: Ablation experiment on the $\lambda$ learning rate on synthetic D-NeRF dataset. The color marks the best and the second best.

| Model | Bouncing Balls | | | Hell Warrior | | | Hook | | | Jumping Jacks | | |
|---|---|---|---|---|---|---|---|---|---|---|---|---|
| | PSNR ↑ | SSIM ↑ | LPIPS↓ | PSNR ↑ | SSIM ↑ | LPIPS↓ | PSNR ↑ | SSIM ↑ | LPIPS↓ | PSNR ↑ | SSIM ↑ | LPIPS↓ |
| Ours (lr=0.0002) | 42.42 | 0.996 | 0.007 | 42.92 | 0.991 | 0.016 | 38.86 | 0.990 | 0.009 | 39.23 | 0.993 | 0.008 |
| Ours (lr=0.002) | 43.32 | 0.996 | 0.007 | 43.13 | 0.991 | 0.014 | 40.18 | 0.992 | 0.008 | 40.52 | 0.994 | 0.006 |
| Ours (lr=0.005) | 42.25 | 0.996 | 0.008 | 42.81 | 0.991 | 0.016 | 39.07 | 0.991 | 0.009 | 38.95 | 0.992 | 0.008 |

| Model | Mutant | | | Standup | | | Trex | | | Average | | |
|---|---|---|---|---|---|---|---|---|---|---|---|---|
| | PSNR ↑ | SSIM ↑ | LPIPS↓ | PSNR ↑ | SSIM ↑ | LPIPS↓ | PSNR ↑ | SSIM ↑ | LPIPS↓ | PSNR ↑ | SSIM ↑ | LPIPS↓ |
| Ours (lr=0.0002) | 43.99 | 0.996 | 0.004 | 46.49 | 0.997 | 0.004 | 40.02 | 0.995 | 0.008 | 41.99 | 0.994 | 0.008 |
| Ours (lr=0.002) | 44.16 | 0.997 | 0.004 | 47.30 | 0.997 | 0.003 | 40.69 | 0.995 | 0.008 | 42.76 | 0.995 | 0.007 |
| Ours (lr=0.005) | 43.89 | 0.996 | 0.005 | 46.50 | 0.997 | 0.004 | 40.66 | 0.995 | 0.008 | 42.02 | 0.994 | 0.008 |

Table 9: Ablation study on the $\lambda$ control range on synthetic D-NeRF dataset. The color marks the best and the second best.

| Model | Bouncing Balls | | | Hell Warrior | | | Hook | | | Jumping Jacks | | |
|---|---|---|---|---|---|---|---|---|---|---|---|---|
| | PSNR ↑ | SSIM ↑ | LPIPS↓ | PSNR ↑ | SSIM ↑ | LPIPS↓ | PSNR ↑ | SSIM ↑ | LPIPS↓ | PSNR ↑ | SSIM ↑ | LPIPS↓ |
| Ours ($0 \leq \lambda \leq 0.5$) | 35.11 | 0.989 | 0.018 | 40.61 | 0.984 | 0.024 | 35.51 | 0.982 | 0.018 | 35.84 | 0.987 | 0.013 |
| Ours ($0 \leq \lambda \leq 1$) | 43.32 | 0.996 | 0.007 | 43.13 | 0.991 | 0.014 | 40.18 | 0.992 | 0.008 | 40.52 | 0.994 | 0.006 |
| Ours ($0.5 \leq \lambda \leq 1$) | 43.17 | 0.996 | 0.008 | 42.70 | 0.990 | 0.018 | 38.64 | 0.989 | 0.010 | 39.00 | 0.992 | 0.009 |

| Model | Mutant | | | Standup | | | Trex | | | Average | | |
|---|---|---|---|---|---|---|---|---|---|---|---|---|
| | PSNR ↑ | SSIM ↑ | LPIPS↓ | PSNR ↑ | SSIM ↑ | LPIPS↓ | PSNR ↑ | SSIM ↑ | LPIPS↓ | PSNR ↑ | SSIM ↑ | LPIPS↓ |
| Ours ($0 \leq \lambda \leq 0.5$) | 38.24 | 0.989 | 0.010 | 40.62 | 0.991 | 0.008 | 35.11 | 0.988 | 0.012 | 37.29 | 0.987 | 0.015 |
| Ours ($0 \leq \lambda \leq 1$) | 44.16 | 0.997 | 0.004 | 46.89 | 0.997 | 0.003 | 40.69 | 0.996 | 0.008 | 42.38 | 0.994 | 0.007 |
| Ours ($0.5 \leq \lambda \leq 1$) | 43.88 | 0.997 | 0.005 | 46.39 | 0.997 | 0.004 | 39.95 | 0.994 | 0.008 | 41.96 | 0.994 | 0.009 |

Furthermore, we constrain the $\lambda$ optimization method, with the rendering results presented in Tab. 9. If $\lambda$ is less than 0.5 and greater than or equal to 0 (corresponding to the right side in Fig. 2, where all Gaussians are low-frequency points), the model struggles to fit high-frequency details, leading to a

significant degradation in reconstruction quality. If $\lambda$ is controlled to be less than or equal to 1 and greater than or equal to 0.5 (corresponding to the left side in Fig. 2, where all Gaussians are high-frequency points), the deformation network's difficulty in regulating the stacking of high-frequency points to fit colors further increases, resulting in poorer reconstruction performance.

Table 10: Ablation study of various embedding types on the D-NeRF dataset. The color marks the best and the second best .

| Model | Bouncing Balls | | | Hell Warrior | | | Hook | | | Jumping Jacks | | |
|---|---|---|---|---|---|---|---|---|---|---|---|---|
| | PSNR ↑ | SSIM ↑ | LPIPS↓ | PSNR ↑ | SSIM ↑ | LPIPS↓ | PSNR ↑ | SSIM ↑ | LPIPS↓ | PSNR ↑ | SSIM ↑ | LPIPS↓ |
| Hash embedding | 31.74 | 0.982 | 0.023 | 33.21 | 0.947 | 0.062 | 26.31 | 0.933 | 0.050 | 27.70 | 0.966 | 0.029 |
| INR embedding | 35.37 | 0.990 | 0.020 | 37.21 | 0.971 | 0.043 | 38.83 | 0.990 | 0.010 | 32.69 | 0.979 | 0.021 |
| Ours | 43.32 | 0.996 | 0.007 | 43.13 | 0.991 | 0.014 | 40.18 | 0.992 | 0.008 | 40.52 | 0.994 | 0.006 |

| Model | Mutant | | | Standup | | | Trex | | | Average | | |
|---|---|---|---|---|---|---|---|---|---|---|---|---|
| | PSNR ↑ | SSIM ↑ | LPIPS↓ | PSNR ↑ | SSIM ↑ | LPIPS↓ | PSNR ↑ | SSIM ↑ | LPIPS↓ | PSNR ↑ | SSIM ↑ | LPIPS↓ |
| Hash embedding | 32.86 | 0.978 | 0.012 | 34.58 | 0.948 | 0.019 | 26.30 | 0.965 | 0.035 | 30.39 | 0.964 | 0.033 |
| INR embedding | 43.25 | 0.996 | 0.005 | 44.89 | 0.996 | 0.004 | 27.37 | 0.970 | 0.032 | 37.09 | 0.985 | 0.019 |
| Ours | 44.16 | 0.997 | 0.004 | 47.30 | 0.997 | 0.003 | 40.69 | 0.995 | 0.008 | 42.76 | 0.995 | 0.007 |

## L  DIFFERENT EMBEDDING COMPARATION

In the FDN, we introduce the High-frequency Fourier Embedding (HFE), which effectively captures high-frequency motion. The low-dimensional visualization in Fig. 13 further demonstrates that the normalized temporal encoding vectors are uniquely distributed without overlaps, indicating a continuous and bijective mapping of time. To further assess the effectiveness of the high-frequency Fourier embedding, we compare it with alternative temporal encoding strategies, including hash-based embeddings and implicit neural representation (INR ) embeddings. Specifically, the hash encoder is implemented with 8 layers, while the INR embedding employs a 3-layer MLP. Quantitative results are reported in Tab. 10, showing that the high-frequency Fourier embedding significantly outperforms the alternatives. This advantage arises from the implicit periodicity in the embedding formulation (Eq. 6), which enables accurate modeling of continuous motion within each period.

## M  ADDITIONAL METRICS

Table 11: Additional metrics comparisons on D-NeRF dataset. The color marks the best and the second best .

| Model | Bouncing Balls | | | Hell Warrior | | | Hook | | | Jumping Jacks | | |
|---|---|---|---|---|---|---|---|---|---|---|---|---|
| | IQA-Q ↑ | IQA-S ↑ | MUSIQ↑ | IQA-Q ↑ | IQA-S ↑ | MUSIQ↑ | IQA-Q ↑ | IQA-S ↑ | MUSIQ↑ | IQA-Q ↑ | IQA-S ↑ | MUSIQ↑ |
| DeformGS | 0.490 | 0.390 | 0.726 | 0.220 | 0.737 | 0.577 | 0.347 | 0.786 | 0.671 | 0.333 | 0.713 | 0.683 |
| 4D-GS | 0.455 | 0.281 | 0.687 | 0.249 | 0.260 | 0.571 | 0.274 | 0.244 | 0.676 | 0.248 | 0.117 | 0.704 |
| SC-GS | 0.625 | 0.555 | 0.703 | 0.231 | 0.772 | 0.600 | 0.370 | 0.722 | 0.675 | 0.375 | 0.740 | 0.687 |
| Grid4D | 0.691 | 0.914 | 0.722 | 0.228 | 0.787 | 0.609 | 0.342 | 0.788 | 0.684 | 0.427 | 0.896 | 0.673 |
| Ours | 0.722 | 0.917 | 0.722 | 0.235 | 0.802 | 0.649 | 0.370 | 0.876 | 0.679 | 0.398 | 0.829 | 0.702 |

| Model | Mutant | | | Standup | | | Trex | | | Average | | |
|---|---|---|---|---|---|---|---|---|---|---|---|---|
| | IQA-Q ↑ | IQA-S ↑ | MUSIQ↑ | IQA-Q ↑ | IQA-S ↑ | MUSIQ↑ | IQA-Q ↑ | IQA-S ↑ | MUSIQ↑ | IQA-Q ↑ | IQA-S ↑ | MUSIQ↑ |
| DeformGS | 0.490 | 0.390 | 0.726 | 0.220 | 0.737 | 0.577 | 0.347 | 0.786 | 0.671 | 0.333 | 0.713 | 0.683 |
| 4D-GS | 0.283 | 0.297 | 0.679 | 0.300 | 0.182 | 0.676 | 0.317 | 0.401 | 0.727 | 0.304 | 0.254 | 0.674 |
| SC-GS | 0.390 | 0.615 | 0.769 | 0.424 | 0.626 | 0.667 | 0.463 | 0.744 | 0.703 | 0.411 | 0.682 | 0.686 |
| Grid4D | 0.432 | 0.902 | 0.674 | 0.427 | 0.896 | 0.673 | 0.580 | 0.960 | 0.712 | 0.447 | 0.877 | 0.678 |
| Ours | 0.467 | 0.898 | 0.684 | 0.437 | 0.895 | 0.680 | 0.560 | 0.961 | 0.712 | 0.452 | 0.883 | 0.690 |

In addition to traditional image-based metrics such as PSNR, SSIM, and LPIPS, we further evaluate the reconstruction quality using model-based assessment methods, including IQA Wang et al. (2022) and MUSIQ Ke et al. (2021). IQA-Q corresponds to the quality dimension, while IQA-S corresponds to sharpness. MUSIQ provides a direct quality score. The quantitative results are summarized in Tab. 11. Compared with the baselines, FAGS consistently achieves higher scores, further demonstrating its superior performance.

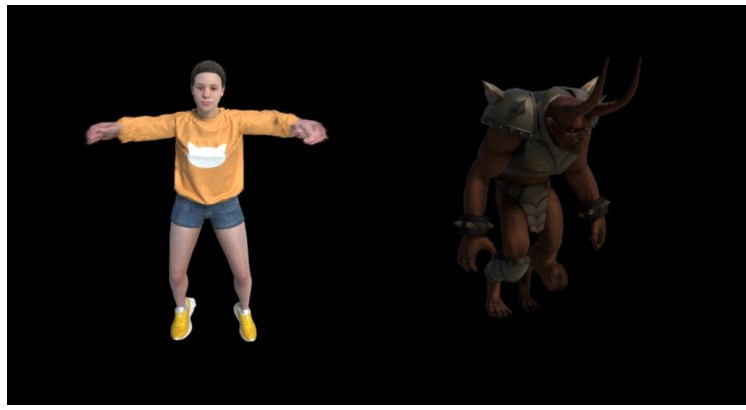

Extrapolate to the 20th cycle

Figure 20: Failure case of extrapolation.

## N    RENDERING SPEED AND COMPUTATIONAL COST

Table 12: Rendering speed comparison on the synthetic D-NeRF dataset.

| Model | Balls | | Warrior | | Hook | | Jumping | | Mutant | | Standup | | Trex | |
|---|---|---|---|---|---|---|---|---|---|---|---|---|---|---|
| | FPS | Num(k) | FPS | Num(k) | FPS | Num(k) | FPS | Num(k) | FPS | Num(k) | FPS | Num(k) | FPS | Num(k) |
| 4D-GS | 182 | 28 | 168 | 40 | 91 | 39 | 207 | 24 | 173 | 38 | 201 | 27 | 151 | 68 |
| DeformGS | 37 | 180 | 161 | 37 | 43 | 150 | 71 | 90 | 49 | 169 | 77 | 81 | 30 | 217 |
| Grid4D | 91 | 192 | 334 | 46 | 79 | 210 | 241 | 68 | 157 | 126 | 170 | 100 | 86 | 254 |
| Ours | 89 | 190 | 289 | 49 | 76 | 221 | 186 | 133 | 123 | 217 | 141 | 122 | 71 | 249 |

**Rendering Speed.** Tab. 12 reports the Gaussian point counts and frames per second (FPS) across different methods on the dataset. Since our approach more effectively identifies high-frequency details, the Gaussian densification rule naturally allocates more Gaussians to represent these regions. This increased number of Gaussians contributes to the superior performance observed in Tab. 1. In terms of rendering speed, the additional Gaussians lead to a slight slowdown compared to Grid4D, but our method still runs significantly faster than DeformGS and 4D-GS. Moreover, due to the precise and efficient modification of the deformation field, the rendering speed reduction relative to Grid4D remains minimal.

Table 13: Computational comparison of different methods on the D-NeRF dataset.

| Model | Training | Inference | VRAM |
|---|---|---|---|
| DeformGS | 14m | 6.0s | 2G |
| 4D-GS | 12m | 2.4s | 2G |
| SC-GS | 56m | 20.7s | 2G |
| Grid4D | 41m | 2.4s | 4G |
| Ours | 55m | 2.8s | 6.6G |

**Time Cost.** We evaluate the time cost of our method and baseline approaches on the D-NeRF dataset. Although our training takes slightly longer and consumes more GPU memory compared to the baselines, FAGS does not incur additional time overhead during testing. At the same time, it achieves the best reconstruction quality.

## O   LIMITATION

**Limitation:** Our method struggles in sparse-view scenarios with wide baselines, where certain points may appear in some views but disappear in others. This inconsistency leads to inaccurate $\alpha$ estimation and erroneous frequency-differentiation behavior. Accurately estimating the frequency-domain $\alpha$ template under sparse-view settings remains a key direction for our future research. Furthermore, when the model is extrapolated beyond the training time, it does not automatically stop but continues to follow the periodic motion, with motion quality gradually decaying. As shown in Fig. 20, after 20 cycles, the local details have noticeably degraded.