# OpenReview forum: "Frequency-aware Dynamic Gaussian Splatting"
_ICLR.cc/2026/Conference — ICLR 2026 Poster_

### Official Review · Reviewer_UoZA · 2025-10-19

**Soundness:** 3
**Presentation:** 3
**Contribution:** 3
**Rating:** 6
**Confidence:** 3

**Summary:**

This paper points out the crucial limitations of previous 4D-reconstruction methods: Most of the previous literatures struggle to balance high-frequency rendering details with high-frequency motion, hence resulting in blurred details. This is mainly due to the limited expressiveness of the 3DGS in the dynamic scenes where the standard alpha-blending of 3DGS fails to express modified geometry along the time steps.

To this end, this paper suggests the novel technique from a frequency perspective where they provide additional learnable parameters into 4DGS pipeline such as $\lambda$, $\beta$ and *frequency-aware gate*, enhancing the expressiveness of 3DGS and successfully distinguishing the low-frequency gaussians and high-frequency gaussians.

Proposed technique contributes to performance improvement to some extent.

**Strengths:**

This paper addresses the meaningful problem where the limited expressiveness of 3DGS fails to recover the fine-grained details across the time steps due to the lack of clear separation between low-frequency gaussians and high-frequency gaussians. This statement is well introduced in the second paragraph of *Introduction part* which makes their motivation clear.

The solution suggested by the paper has novelty inside where they improve the expressiveness of 3DGS by introducing the additional learnable parameters from a frequency perspective, clearly separating the gaussians into high-frequency GS and low-frequency GS. Furthermore, ablation studies support the effectiveness of each part where Tab. 4 shows the quantitative improvements and Fig. 7 demonstrates the clear separation of the gaussians into low-frequency and high-frequency components.

**Weaknesses:**

1. **Marginal performance improvement**

* It seems like the performance improvement in SSIM and LPIPIS is quite marginal (mostly +0.001 in both metrics) compared to the Grid4D while performance in PSNR is relatively clear. Furthermore, in *Trex* scene of Tab. 1, it seems that SC-GS achieves the best result in LPIPS metric while the paper indicates 4DGS and their own method achieves the best scores.
* Why is LPIPS metric not reported in Neu3D dataset? Also, please report the average scores in Neu3D dataset.
* In Fig. 6, I failed to find the clear improvement  between Grid4D vs paper's methods, especially in 2nd and 3rd rows. In 2nd row, I think the both of Grid4D and paper's method fail to reconstruct the clear details.
* The combination of 3 rendering matrices compared to GT (PSNR, SSIM, LPIPS) is sometimes not sufficient to show the effectiveness of the method. It would be better to adopt additional metrics such as MUSIQ [a] and CLIP-IQA [b] to estimate the naturalness of the images since the paper points out the blurred reconstructed details from the previous methods.

2. **Room of improvement in ablation study**

* In Sec. 3.2, paper introduces two learnable parameters, $\lambda$ and $\beta$, in the alpha-blending part of the 3DGS to improve the expressiveness of the 3DGS. $\lambda$ controls the transparency gradient from the projection center, effectively separating the high-frequency GS and low-frequency GS, while $\beta$ additionally controls the differentiated regions. What will be happened if we remove the $\beta$? It would be better if the paper reports the performance when they only use one of them ($\lambda$ and $\beta$).
* Tab. 4 presents the ablation study for each component of FAGS. However, more diverse combinations should be explored.

Considering the contributions and weaknesses of this paper, I would like to recommend the score between 6 to 8 which is around 7.

---

**References**

[a] Junjie Ke, Qifei Wang, Yilin Wang, Peyman Milanfar, and Feng Yang. Musiq: Multi-scale image quality transformer. In Proceedings of the IEEE/CVF international conference on computer vision, pages 5148–5157, 2021.

[b] Jianyi Wang, Kelvin CK Chan, and Chen Change Loy. Exploring clip for assessing the look and feel of images. In Proceedings of the AAAI Conference on Artificial Intelligence, pages 2555–2563, 2023

**Questions:**

**Primary area**: Why is the primary area of this paper indicated as *generative model*?

---

> ### Author Response · Authors · 2025-11-23
>
> We thank the reviewer for the suggestion. In response, we have provided more detailed experimental results in both the main text and appendix, and **we recommend consulting the updated manuscript.**
>
> **W1: SSIM and LPIPIS is quite marginal. SC-GS appears to achieve the best results in the Trex.**
>
> Following the explanation in [1], PSNR evaluates reconstruction quality at the pixel level, SSIM at the structural level, and LPIPS at the semantic level. Because our method resolves the conflict between high-frequency details and high-frequency motion—an issue that primarily influences fine-grained appearance—it naturally yields larger gains in pixel-level metrics. The observed improvement in PSNR therefore substantiates the effectiveness of our approach. We also note that the previously reported LPIPS score for SC-GS was a typographical error and has now been corrected.
>
> [1]  Semantic Similarity Score for Measuring Visual Similarity at Semantic Level
>
> **W2: Need LPIPS and average scores on Neu3D**
>
>
>
> We report average values in the table below, where our FAGS outperforms the baseline on LPIPS. Complete results are provided in Tab. 2.
> |Metric|PSNR↑|SSIM↑|LPIPS↓
> |-|-|-|-
> |4D-GS|30.65|0.934|0.161
> |Grid4D|31.63|0.937|0.149
> |Ours|32.18|0.946|0.146
>
>
>
>
> **W3: I failed to find the clear improvement in Fig.6**
>
> We have re-annotated Fig. 6 to emphasize the key differences. In the highlighted regions, FAGS produces clearer text, more sharply defined knife-tip structures, and reduced blur compared to Grid4D. Notably, the example in the last row further demonstrates FAGS’ substantial improvement in preserving fine details.
>
>
> **W4：Need introduce more metrics such as MUSIQ and CLIP-IQA**
>
> In Tab.11, we additionally report MUSIQ, IQA-Q (quality), and IQA-S (sharpness). Across most scenarios, FAGS achieves the highest scores among all baselines and also ranks first in terms of average performance.
>
> |Metric|IQA-Q ↑|IQA-S ↑|MUSIQ↑
> |-|-|-|-
> |DeformGS|0.333|0.713|0.683
> |4D-GS|0.304|0.254|0.674
> |SC-GS|0.411|0.682|0.686
> |Grid4D|0.447|0.877|0.678
> |Ours|0.452|0.883|0.690
>
>
>
>
> **W5: It would be better if the paper reports the performance when they only use one of them (α and β).**
>
>
> We add ablation studies in Tab.4 for the cases of removing only α and only β, and we report the averages below.
> |Metric|Full|w/o FDGK.λ|w/o FDGK.β
> |-|-|-|-
> |PSNR↑|42.76|42.30|42.26
> |SSIM↑|0.995|0.994|0.994
> |LPIPS↓|0.007|0.008|0.008
>
>
>
>
> **W6：Tab.4 presents the ablation study for each component of FAGS. However, more diverse combinations should be explored.**
>
> We have added the results of removing both FG and $L_{fre}$ in Tab.4. These results further demonstrates that FG and $L_{fre}$ play complementary roles in FAGS and are essential for 4D reconstruction.
>
> |Metric|Full|w/o (FG+$L_{fre}$)
> |-|-|-
> |PSNR↑|42.76|42.43
> |SSIM↑|0.995|0.994
> |LPIPS↓|0.007|0.008

---

### Official Review · Reviewer_cavT · 2025-10-23

**Soundness:** 2
**Presentation:** 1
**Contribution:** 2
**Rating:** 4
**Confidence:** 4

**Summary:**

The authors propose Frequency-Aware Dynamic Gaussian Splatting, or FAGS. FAGS composes of several components: it first intrudes FDGK, which plays on modifying the Gaussian distribution of a GS primitive. FDGK divides the Gaussian distribution into three regions, and modulate the scale and bias of each region, such that the distribution is more peaky or flatter based on optimization. It then introduces additional encoding scheme for the temporal domain, i.e. encoding time into higher dimensions/frequency bands compared to the spatial domain, such that the MLP can disentangle motion better. Frequency gating and a loss in frequency domain is introduced, though their effectiveness is not very evident. Overall, FAGS outperforms previous approach in terms of standard metrics; however, it is unclear to me if this is due to better disentanglement of high frequency detail and motion, or simply a better Gaussian representation capacity.

**Strengths:**

1. FAGS outperforms existing methods on standard 4D reconstruction datasets, particularly on synthetic datasets. Real dataset performances have a smaller improvement, but that seems reasonable given potential noise in real datasets.
2. The FDGK formulation does seem to improve Gaussian representation capacity and leads to better fitting.

**Weaknesses:**

I have several issues with this paper, specifically concerning the writing (difficult to read, bold claims without evidence, etc.) and the necessity of multiple parts. I think overall the paper can be more focused on FDGK and better distinguish that contribution.

1. It is hard to follow this work, multiple variables are not defined. E.g. in Eq. (2), g is not defined. D is not defined in Eq. (8), multiple abuse of notation, e.g., where "MLP" indicates multiple neural networks with different parameters. The math is unclear, especially Eq. (2). How is this equation related to "numerical fluctuations"? While the first part of the Equation makes sense, the second part, where r is said to be able to go to infinity, does not seem mathematically correct. We may be able to scale with r in limited region of the distribution, by certainly not the entire distribution, otherwise opacity can be scaled to infinity. Proper constrains should be stated. I have to reference prior work, e.g., DRK to double-check my understanding. The details on frequency-aware gate score is very sparse, it is unclear to me why this score can do the things that is claimed. All that is conveyed is that this score seems to come out of a neural network.

2. Multiple claims seem to be overboard/abusing concepts that have similar words but not really related. The main claim: "FAGS can disentangle high freq. detail and high freq. motion" does not feel properly addressed by FDGK. The FDN introduces higher dimensional embedding to the temporal axis with Fourier basis - I am not sure if this is necessarily related to motion frequency rather than higher dimensional embedding leads to better performance. Similarly, it is unclear to me how the computing the amplitude spectra loss over an image leads to better frequency disentanglement; both FG and L_fre seem to contribute very little based on ablation.

3. It is unclear to me why DRK is not a baseline at least in the ablation study. It seems like this work is closely related to DRK. Despite that DRK does not explicitly address 4D reconstruction, it should be straightforward to add DRK as a kernel representation without affecting the motion estimation network too much. Without such experiments, it is unclear to me if FDGK is a simpler variant of DRK with pre-defined regions of Gaussians to optimize.

I overall think this paper has interesting contributions, and would give a borderline accept if FDGK/high frequency detail and motion disentanglement can be more grounded, which may require datasets that have some notion of what "high/low frequency motions" are. Given the current state of writing, and the (seemingly) unnecessary components, I vote for borderline reject and think the authors should significantly modify the writing.

**Questions:**

1. WRT the main claim: other than visual quality, is there anything else that proves the predicted motion is now "higher frequency" and more accurate, e.g., what is the "correct" motion frequency? My impression is that FDGK does improve the representation capacity of individual Gaussians and does improve visual performance; however, its relationship to motion is unclear. I.e., if FDGK is applied to standard 3D reconstruction, will it not improve performance?
2. Higher dimensional embedding to the temporal axis with Fourier basis: is this related to anything specific about Fourier space, or alternative embedding scheme, e.g. higher dimension hash embedding other implicit neural representation embedding, would achieve similar results?
3. Are FG and L_fre necessary? Why can frequency-aware gate score behave in the manner that is described? Based on ablation, FG and L_fre contributes very minimally.

---

> ### Author Response · Authors · 2025-11-23
>
> We thank the reviewer for the suggestions. We have revised the manuscript to clarify ambiguous definitions and added extensive experiments in the main text and appendix to demonstrate FAGS’ high-frequency motion capture capabilities. **We encourage the reviewer to consult the updated manuscript.**
>
> **W1: How Eq(2) is related to "numerical fluctuations". Need check definition.**
>
>
>
> - In Eq.2, **$g = \exp(-\frac{1}{2} (p - \mu_i^{2D})^T (\Sigma_i^{2D})^{-1} (p - \mu_i^{2D}))$** causes numerical fluctuations, as it rapidly affects the α of a Gaussian on the image plane depending on the distance between a pixel **p** and the Gaussian center. To clarify Eq.2 further, we have added additional constraints: $\alpha_i =\mathrm{min}( o_i \psi(g),0.99)$, where 0.99 is the truncation threshold used in GS rendering.
>
>
> **W2/Q1/Q3: It is unclear whether FAGS truly disentangles high-frequency detail from high-frequency motion, and the necessity and impact of FG and L_fre appear minimal.**
>
>
> We analyze the motion trajectories of all Gaussians, quantifying and visualizing the temporal motion power spectrum distribution and temporal second-order finite-difference energy heatmaps to demonstrate the high-frequency motions learned by FAGS. Additionally, we provide ablation studies confirming the necessity of FG and $L_{fre}$.
>
>
> -  Power spectrum (Fig.11, Appx.Fig.14)
> FAGS shows a right-shifted and taller spectrum compared to the baseline, indicating more Gaussians contributing to high-frequency motion and better preservation of rapid dynamics that baselines oversmooth.
> - Finite-difference energy heatmap (Appx.Fig.15)
> With time on the horizontal axis and Gaussian index on the vertical, FAGS produces sharper, higher-contrast patterns, demonstrating that FDN accurately captures localized high-frequency motion rather than diffusing it across frames.
> - FG: Controls how strongly the deformation network updates Gaussian parameters over time, enabling motion-level frequency control beyond FDGK (Fig. 16).
> - $L_{fre}$: Captures subtle high-frequency color details and guides FDN to control FDGK across multiple time steps, especially for faster edge motion (Fig. 8).
>
> |Metric|Full|w/o (FG+$L_{fre}$)
> |-|-|-
> |PSNR↑|42.76|42.43
> |SSIM↑|0.995|0.994
> |LPIPS↓|0.007|0.008
> - Above table ( also in Tab. 4) shows that removing FG and $L_{fre}$ significantly reduces PSNR, SSIM, and LPIPS, confirming their key role in modeling high-frequency motion and fine-grained details.
>
>
>
> **W2/Q2: Higher dimension hash embedding and other implicit neural representation embedding, would achieve similar results?**
>
>
> We provide a comparison of high-frequency motion modeling using HFE versus other embeddings; complete metrics are reported in Tab.10.
>
>
> |Model|Ball|Hell|Hook|Jacks|Mutant|Standup|Trex|Avg.
> |-|-|-|-|-|-|-|-|-
> |Hash|31.74|33.21|26.31|27.70|32.86|34.58|26.30|30.39
> |INR|35.37|37.21|38.83|32.69|43.25|44.89|27.37|37.09
> |Ours|43.32|43.13|40.18|40.52|44.16|47.30|40.69|42.76
>
>
>
> INR: implicit neural representation. We report PSNR (higher is better) here.
>
> The above table (with additional metrics in Tab.~10) shows that other embeddings underperform compared to our HFE, as higher-dimensional hash embeddings and implicit neural representations both lack frequency awareness and introduce parameters that are difficult to optimize.
>
>
>
> **W3: Need comparison with DRK**
>
>
>
> We compared FDGK and DRK in both 4D and 3D scenes. Quantitative (Tab.1&6) and qualitative (Fig.4&9) results demonstrate the advantages of FDGK over DRK.
>
> |4D|PSNR↑|SSIM↑|LPIPS↓
> |-|-|-|-
> |Grid4D+DRK|39.43|0.990|0.015
> |ours|42.76|0.995|0.007
>
> |3D|PSNR↑|SSIM↑|LPIPS↓
> |-|-|-|-
> |3DGS|28.20|0.24|0.93
> |DRK|28.15|0.24|0.92
> |FDGK|28.32|0.26|0.92
>
> - Extensive 4D/3D comparisons are provided (Tab. 1 & 6, Fig. 4 & 9); FDGK achieves advantages on several metrics, particularly in 4D scenes.
> - 4D: DRK’s flexible kernels can hinder Gaussian control and sometimes degrade reconstruction; FDGK differentiates early and remains stable (Tab.1 & Fig.4).
> - 3D: FDGK outperforms DRK on some metrics, showing partial advantages while remaining competitive (Tab.6 & Fig.9).

---

### Official Review · Reviewer_chQR · 2025-10-27

**Soundness:** 3
**Presentation:** 2
**Contribution:** 3
**Rating:** 6
**Confidence:** 3

**Summary:**

The paper proposes Frequency-Aware Dynamic Gaussian Splatting, a 4D Gaussian-splatting framework aimed at reducing motion blur in dynamic novel-view synthesis. The authors argue that prior deformation-field-based dynamic Gaussian methods struggle with a *spectral conflict*: they must both (i) stack many soft Gaussians to reproduce high-frequency spatial details, and (ii) drive those same Gaussians along accurate, potentially fast and highly non-smooth temporal trajectories. As a result, the deformation network tends to bias toward globally smooth, low-frequency motion, which causes blurred boundaries and smeared fast-moving parts under novel viewpoints.

To address this, the paper introduces two technical components.

- Frequency-Differentiated Gaussian Kernel (FDGK): Instead of using a fixed isotropic alpha falloff per Gaussian, each Gaussian gets a learnable piecewise alpha modulation ψ(g) with parameters λ (controls sharpness vs smoothness of the opacity profile) and β (controls the spatial extent / boundary of that sharpness). Gaussians can thus specialize as high-frequency (sharp boundaries, for edges) or low-frequency (smooth support for broad regions), reducing the need to stack many Gaussians to recover detail and easing pressure on the deformation network.

- Fourier-Deformation Network (FDN): The deformation network fuses low-frequency spatiotemporal hash encodings with high-frequency temporal Fourier embeddings, whose amplitudes are predicted per Gaussian. A frequency-aware gate η modulates how strongly each Gaussian is allowed to move at each time step, suppressing motion in static/background regions while permitting rapid, localized motion where needed. A Fourier-domain loss on amplitude spectra further emphasizes fitting high-frequency components.

Experiments on D-NeRF, HyperNeRF, and Neu3D show consistent improvements in PSNR / SSIM / LPIPS / MS-SSIM over strong baselines such as 4D-GS, DeformGS, SC-GS, and Grid4D, especially around fast-moving fine structures (fingers, tools, flames). Visualizations of per-Gaussian motion trajectories and ablations suggest that FDGK + FDN reduce drifting / smearing and yield sharper reconstruction of both spatial detail and rapid motion.

**Strengths:**

- Clear motivation (motion blur in dynamic GS): The paper focuses on a highly visible failure mode in existing dynamic Gaussian splatting pipelines: smeared or drifting fine details (e.g., fingertips, blades, flames) in novel-view renderings. This is both practically important and easy to judge qualitatively.
- Theoretically solid contributions: Frequency-Differentiated Gaussian Kernel (FDGK) allows each Gaussian to learn its own alpha profile via λ (sharp vs smooth) and β (boundary span), which lets some Gaussians specialize in high-frequency edges without needing dense stacking, while others handle smooth regions. Fourier-Deformation Network injects explicit high-frequency temporal bases and modulates each Gaussian’s deformation strength with a learned gate η, which is a reasonable mechanism to retain fast, local, non-smooth motion for moving parts while not injecting jitter into the static background.
- Analysis of parameter distributions: The supplementary shows how λ splits Gaussians into high-frequency vs low-frequency groups during training, and how β adapts boundary spans. The gate η distribution highlights that not all Gaussians become fully dynamic. This supports the claim that the method self-organizes a frequency-aware decomposition.

**Weaknesses:**

- Core causal story is qualitative, not rigorously established.
The paper’s central narrative is that existing methods blur because they cannot jointly model high-frequency spatial detail and high-frequency motion, and that FDGK+FDN resolves this spectral conflict. While the visual evidence is compelling, there is no direct quantitative analysis of temporal frequency content (e.g., Fourier spectrum of per-Gaussian trajectories before/after, or a measure of per-point temporal curvature). Maybe Fig.10 tries to display this point, but there is no sufficient explanation of Fig.10. The argument remains largely anecdotally supported via qualitative trajectory plots and better metrics.
- Definition of “high-frequency motion” is not clear.
The method heavily leans on the dichotomy of “low-frequency global motion” vs “high-frequency local motion,” but never formalizes it (e.g., in terms of time derivatives or spectral energy bands), even though it motivates the design of FDGK, the Fourier embedding, and the gating mechanism. A clearer mathematical definition would make the framing more than just intuitive storytelling.
- Fourier-Deformation Network assumes a quasi-periodic temporal structure.
Modeling trajectories as Fourier bases with learned amplitudes implicitly assumes motions can be expressed as superpositions of (possibly periodic) sinusoids. From the videos in supplementary materials, most the the shown case can be figured out certain periodic motion pattern. Real-world motion like sudden stops, impacts, or non-repetitive gestures may violate this assumption. The paper does not explore failure cases or quantify when Fourier modes become insufficient.
- Clarity / naming / polish.
The paper alternates between **FAGS** and **FDGS**. Fig.10 lacks an explanation. The gating equation (Formula 8) does not have a sufficient explanation. These presentation issues hurt readability.

**Questions:**

These are questions where a rebuttal could meaningfully improve my score:

- Can you quantitatively define/measure high-frequency motion?
For example, could you report, for baseline vs. your method, the temporal power spectrum or second-order temporal finite-difference energy of a set of tracked Gaussians on fast-moving parts (e.g., fingertips in JumpingJacks)? Right now, high-frequency motion is central but informal.

- Can you give a more concrete analysis supporting the “spectral conflict” claim?
In Fig.12, you show that baseline Gaussians drift to other regions to satisfy both detail fitting and motion alignment, which you argue leads to blur. Could you quantify how often this cross-region drift happens, and how it correlates with blur artifacts in novel views? This would turn the nice qualitative story into measurable evidence.

---

> ### Author Response · Authors · 2025-11-23
>
> We thank the reviewer for the insightful suggestion, which led us to investigate metrics of high-frequency motion  and the trade-off between high-frequency details and motion. **These updates have been incorporated into the main paper, and we strongly recommend reviewing the revised manuscript for the changes.**
>
>
> **W1/Q1: Need quantitative analysis of temporal frequency content**
>
>
>
> **Detailed analysis is provided in Appx.C.**
> In Fig.11 and Appx. Figs.14–15, we show quantitative and visual evidence of the temporal-frequency behavior of all Gaussians, including their temporal power spectrum and second-order finite-difference energy.
>
> - **Power spectrum (Fig.11, Appx.Fig.14)**
>     A right-shifted spectrum indicates stronger high-frequency motion, while taller bands show that more Gaussians contribute to these components. FAGS consistently exhibits a right-shifted distribution compared with the baseline, demonstrating that it preserves and models high-frequency motions that baselines tend to oversmooth.
>
> - **Finite-difference energy heatmap (Appx.Fig.15)**
>     Time is on the horizontal axis and Gaussian index on the vertical axis; brighter regions correspond to stronger motion. FAGS produces sharper, higher-contrast patterns, confirming that FDN captures localized high-frequency motion at the correct moments rather than diffusing it across frames.
>
> Overall, these results highlight that FAGS more faithfully maintains high-frequency temporal dynamics and models localized motion better than Grid4D.
>
>
>
> **W2/Q2: Can you give a more concrete analysis supporting the “spectral conflict” claim? Could you quantify how often this cross-region drift happens, and how it correlates with blur artifacts in novel views?**
>
>
> We propose Conflict Risk to quantitatively measure “spectral conflict” (see the table below) and provide detailed analysis in Appx.D
>
> |Model|Ball|Hell|Hook|Jacks|Mutant|Standup|Trex|Avg.
> |-|-|-|-|-|-|-|-|-
> |Grid4D|15.3%|69.1%|42.0%|36.9%|44.9 %|41.0%|35.2%|40.6%
> |Ours|9.8%|36.2%|33.2%|11.9%|29.7%|31.2%|29.0%|25.1%
>
>
>
> - **Conflict Risk** measures how often Gaussians exhibit drifts that induce spectral conflict. We compute per-Gaussian gradients on test views without updating either attributes or the deformation network, and define Conflict Risk as the proportion of Gaussians whose gradient magnitude exceeds a pruning threshold.
>
> - **Analysis:** A high Conflict Risk indicates that Gaussians still produce large gradients in novel views despite being fully optimized in training views, suggesting that the deformation network overfits high-frequency appearance rather than motion. As reported in Tab.5, lower Conflict Risk correlates with higher PSNR and SSIM and lower LPIPS. Our method substantially reduces Conflict Risk compared with the baseline, mitigating blur and other artifacts.
>
>
> **W3: The paper does not explore failure cases when Fourier modes become insufficient**
>
>
> In Appx. Sec.O, we discuss failure cases. To explore when Fourier modes become insufficient, we compress the training time range from [0,1.0] to [0,0.5], simulating abrupt stops or non-repetitive motions. Beyond the training range, the model continues periodic motion, but reconstruction quality gradually decays. As shown in Fig.~20, local details noticeably degrade after approximately 20 cycles.
>
> **W4：Need to check typos and improve clarity.**
>
>
> Thank you for the valuable suggestion. We have unified the terminology by correcting FDGS to FAGS, clarified the description of Fig.10, and improved the explanation of Eq.8 to enhance readability.

---

> ### Comment · Reviewer_chQR · 2025-11-27
>
> Thanks for the rebuttal. It resolves most of my concerns but further confirms that this method will be limited by a quasi-periodic temporal structure. I will keep my rating unchanged.
>
> Besides, the author still needs to check typos ("correcting FAGS to FAGS" in W4 reply). The typo in the reply to typos makes it somewhat funny.

---

### Official Review · Reviewer_77AF · 2025-10-30

**Soundness:** 3
**Presentation:** 2
**Contribution:** 3
**Rating:** 6
**Confidence:** 3

**Summary:**

The authors attribute motion blur in prior dynamic Gaussian splatting methods to the inherent difficulty of simultaneously modeling high-frequency rendering details and high-frequency motion. To address this issue, they propose a deformation-field-driven framework that explicitly separates Gaussians into low- and high-frequency components, allowing each group to specialize in reconstructing its respective part before merging. Central to this framework is a Fourier-Deformation Network, which leverages Fourier embeddings to effectively capture fine-grained high-frequency local deformations while suppressing redundant motion in low-frequency regions. In addition, the method introduces a frequency-differentiated Gaussian kernel that adaptively sharpens or smooths Gaussians via learnable parameters, enabling clear separation between high- and low-frequency behavior. Extensive experiments on dynamic scene datasets demonstrate the effectiveness of the approach, supported by thorough ablation studies that validate each proposed component.

**Strengths:**

1. The paper proposes a deformation-field-driven method that is compatible with most existing dynamic 3D Gaussian Splatting approaches and it has a good theoretical explanation of solving motion blur.
2. The introduction of a frequency-aware deformation field provides a clear and effective strategy for mitigating motion blur in dynamic scenes, especially under novel views.
3. Experiments demonstrate that the proposed method reduces motion blur compared to prior dynamic Gaussian Splatting methods.

**Weaknesses:**

1. The paper lacks a clear analysis of the computational overhead introduced by the frequency-aware module, such as runtime or memory usage, especially in comparison to existing baseline methods (In your pipeline, it has 4 MLPs. I think it's necessary to show the training time and evaluation time in your paper).
2. Although the approach shows that it reduces motion blur with normal input, the paper does not include comparisons with existing approaches that mainly address deblurring. Moreover, it does not evaluate on datasets with motion blur to validate its effectiveness under such conditions.
3. The description of the formulas are not so clear, e.g. you should clarify the 'g' in expression(2)~(4).

**Questions:**

1. Could you please offer me the training time and evaluation time of the baselines in paper? And could you please tell me how many layers in your MLPs?
2. Belowing current papers are focusing on dynamic, could you please evaluate your model on blurring dataset to see whether it can reduce heavy motion blur?
3. Could you select one baseline from the following list of papers and then compare both time metrics and result metrics between that baseline and your approach?


[1]. BARD-GS: Blur-Aware Reconstruction of Dynamic Scenes via Gaussian Splatting (Deblurring Method + Blurring Dataset)

[2]. Deblur4DGS: 4D Gaussian Splatting from Blurry Monocular Video (Deblurring Method)

[3]. MoBGS: Motion Deblurring Dynamic 3D Gaussian Splatting for Blurry Monocular Video (Deblurring Method)

---

> ### Author Response · Authors · 2025-11-23
>
> Thank you for the reviewer’s helpful suggestion. Incorporating additional datasets and model comparisons further highlights the advantages of our method in extreme motion-blur scenes. **We have updated the main paper accordingly, and we strongly recommend reviewing the revised manuscript first.**
>
> **W1/Q1: Need more computational overhead. Show the training time and evaluation time. How many layers in your MLPs.**
>
>
> We provide detailed computational cost data and clarify the definition of “MLPs.”
> |Model|Training|Inference|VRAM
> |-|-|-|-
> |DeformGS|14m|6.0s|2G
> |4D-GS|12m|2.4s|2G
> |SC-GS|56m|20.7s|2G
> |Grid4D|41m|2.4s|4G
> |Ours|55m|2.8s|6.6G
> - We have reported the runtime and VRAM usage in above table (also in Appx.Tab.13) Under configurations that achieve SOTA performance, our training time is comparable to SC-GS, and the inference time is similar to Grid4D. The VRAM usage is relatively higher, mainly because FAGS requires a larger number of trainable parameters.
> - Each ‘MLPs’ consists of one linear layer followed by a ReLU layer, as presented in Appx.Fig.12. To prevent misunderstanding, we have refined and clarified Fig.3.
>
>
> **W2/Q2/Q3: Need include comparisons with existing approaches that mainly address deblurring. Evaluate FAGS on blurring dataset.**
>
>
> FAGS focuses on handling the general blurring that still occurs with normal inputs. Following your advice, we compared our method with the SOTA deblurring approach BARD-GS on the motion-blur dataset purposed in their paper. The quantitative results are reported in Appx.Tab.7, and the qualitative results are shown in Fig.10.
>
> |Model|PSNR↑|SSIM↓|LPIPS↓|Time↓
> |-|-|-|-|-
> |DyBluRF$^*$|20.57|0.761|0.361|-
> |DeformGS$^*$|22.83|0.824|0.350|-
> |DeformGS + MPRNet$^*$|23.10|0.836|0.263|-
> |4DGS$^*$|22.06|0.809|0.373|-
> |4DGS+MPRNet$^*$|22.58|0.829|0.286|-
> |BARD-GS$^*$|25.13|0.851|0.156|-
> |BARD-GS#|23.46|0.847|0.183|380m
> |Ours|23.12|0.839|0.201|**45m**
> |Ours+MPRNet|**25.42**|**0.873**|**0.135**|50m
>
> $*$ indicates the results reported by BARD-GS, and # denotes our reproduced results.
>
> FAGS outperforms other baselines without requiring MPRNet pre-deblurring (above, Appx.F), and achieves SOTA performance with MPRNet. It shows clear advantages in edge preservation under motion blur (Fig. 10). These results demonstrate that FAGS remains effective under severe motion blur, primarily because FDGK separates high- and low-frequency components for better detail modeling, and FDN precisely controls Gaussian with different motion frequency.
>
> **W3: Should clarify the g in expression(2)~(4).**
>
> We have provided a detailed explanation regarding Eq. 2~4 in our paper.
> $g = \exp(-\frac{1}{2} (p - \mu_i^{2D})^T (\Sigma_i^{2D})^{-1} (p - \mu_i^{2D}))$

---

> ### Comment · Reviewer_77AF · 2025-11-27
>
> Thank you for providing the additional experiments on runtime and GPU consumption, which resolves my concerns. I also appreciate the analysis conducted under blurred input conditions. It is encouraging to observe that your pipeline not only achieves better performance than prior methods in this setting but does so with significantly reduced computational cost, demonstrating both robustness and efficiency.
>
> Therefore, I decide to raise my score.

---

### Author Response · Authors · 2025-11-27
**Follow-up on Rebuttal – Paper 3105**

Dear reviewers,

We hope this message finds you well. As the discussion phase nears its end, we would greatly appreciate it if you could briefly review our response and revised paper at your convenience. We are happy to clarify any questions.

Thank you for your time and valuable feedback.

Best regards,
Authors of Paper 3105

---

### Author Response · Authors · 2025-12-01
**Summary for Area Chair**

Dear Area Chair,

We fully understand the heavy workload during this critical decision period and sincerely appreciate the time and dedication you and the reviewers devote to maintaining high review standards.

To support your final decision, we provide a concise summary of the paper’s key contributions and the consensus reached during the rebuttal phase.

**Paper Overview & Consensus:**

We propose **FAGS**, a novel 4D reconstruction method that resolves blur by addressing the **spectral conflict between high-frequency details and high-frequency motion**, achieving SOTA results. Reviewers highlight the novelty and effectiveness of this frequency-based perspective. In the rebuttal, we added experiments on extreme motion blur, quantified motion frequency, and analyzed spectral conflict. Before score rollback, Reviewers 77AF and chQR confirmed that their key concerns were addressed, reflecting a consensus toward acceptance.

**Initial Scores: 6,6,4,6**

**Recognized Strengths:**

* FAGS is universally affirmed to **solve the critical problem** of smeared fine details and motion blur inherent in dynamic GS novel-view synthesis.
* Our Frequency-Differentiated Gaussian Kernel and frequency-aware deformation network are recognized as a **theoretically solid and novel strategy** for actively separating high- and low-frequency motion.
* Experimental results confirm **FAGS’s superior performance** and effective frequency-aware decomposition.

**Score Changes & Discussion Highlights**

* **77AF: 6→8 (Accept):** The initial evaluation is positive. After the rebuttal, the reviewer concluded: _"...your pipeline not only achieves better performance than prior methods ... with significantly reduced computational cost, demonstrating both robustness and efficiency. ... **I decide to raise my score.**"_
* **chQR: 6:** Initially positive. After we provided **quantitative results, analyses, and comparisons demonstrating FAGS’s ability to capture high-frequency motion and reduce spectral conflict**: _"It resolves most of my concerns... I will keep my rating unchanged."_
* **cavT: 4:** Finds FAGS interesting but needs evidence that the **improvement comes from modeling high-frequency motion rather than from FDGK’s general gains**.  (a) **Broader comparison:**  We compare FAGS with DRK + Grid4D in 4D scenes and with DRK alone in 3D scenes. Results show that the **4D gains arise from the synergy of FDGK and FDN**, with FDGK also achieving higher PSNR than DRK in 3D scenes. (b) **Quantitative evidence:** Tracking Gaussian-point motion shows  **FDN captures more high-frequency motion than the SOTA baseline**.
*  **UoZA: 6 (_around 7_):** Positive initial review, acknowledging the significance of our work. During the rebuttal, we added **extensive quantitative metrics** (IQA-Q, IQA-S, MUSIQ) and **fine-grained ablations** (w/o FDGK.λ, w/o FDGK.β, w/o (FG + $L_{fre}$)), further strengthening evidence of FAGS’s effectiveness.

**Main Concerns Resolved**

|**Concern**|**Resolution**|
|-|-|
|**Comparison on Motion Blur Dataset** (77AF)|Qualitative (Fig.10) and quantitative (Tab.7) results show **FAGS outperforms SOTA baseline** (e.g., BARD-GS), even under **severe motion blur**.|
|**Quantifying Spectral Conflict** (chQR)|**Introduce a conflict risk metric** and **quantitative analysis** **of spectral conflict**, demonstrating that **reducing conflict risk correlates with better novel-view quality**  (Tab.5).|
|**Measuring High-Frequency Motion** (chQR, cavT)|The motions of Gaussians are **quantified using Temporal Power Spectrum distributions and Finite-Difference Energy heatmaps**, demonstrating that **FAGS captures more high-frequency motion than the baselines** (Fig.14–15).|
|**Gains from disentanglement vs. kernel** (cavT)| Results (Fig.4, Tab.1) show that using a flexible 3D Gaussian kernel (e.g., DRK) in place of 3DGS within Grid4D (SOTA baseline) does not naturally improve 4D reconstruction. Our SOTA results demonstrate the **gains  from synergy of FDGK (high-frequency detail) and FDN (high-frequency motion)**. 3D results (Fig.10, Tab.6) confirm FDGK outperforms DRK in PSNR. |
|**Component Necessity** (cavT)| Qualitative (Fig.7, Fig.8, Fig.16) and quantitative (Tab.4, Tab.10) results **validate the necessity of the Gate and $L_{fre}$** and show that our **HFE surpasses other temporal embeddings**. |
|**Evaluation with More Metrics** (77AF, UoZA)|**Evaluation with IQA-Q, IQA-S and MUSIQ** confirms that **FAGS outperforms the baselines** (Tab.11). We report computational overhead (Tab.7, Tab.13). |

We believe our detailed rebuttal and the extensive additional experiments (including new baselines, spectral conflict and high-frequency motion quantification, DRK comparisons, and detailed ablations) have thoroughly resolved all reviewers' initial queries. We hope this summary assists you in your final decision.

Thank you once again for your service to the community.

With sincere gratitude,

The Authors of FAGS.

---

### Meta-Review · Area_Chair_u3io · 2025-12-31

**Summary:**

The paper proposes FAGS, a 4D Gaussian-splatting framework designed to reduce blur in reconstructions, which is a known issue in dynamic Gaussian Splatting. The authors attribute this blur to *spectral conflict* between high-frequency details and high-frequency motion. To address this, the work enhances the expressiveness of Gaussian Splatting using FDGK, which modifies the opacity decay with distance from the Gaussian center, and introduces a new deformation field architecture that incorporates Fourier features.

In the initial reviews, three reviewers gave weak acceptance and one gave weak rejection. After the rebuttal, one of the weak accept reviewers indicated a score increase to 8. Overall, there is consensus that the paper presents interesting and theoretically grounded contributions, offering a novel strategy for separating high- and low-frequency motion. The experimental results further support the effectiveness of FAGS, showing strong performance even under *blurred input conditions*.

Reviewers initially raised a few weaknesses, but the consensus is that the authors have addressed most of them effectively in the rebuttal.
I generally agree with reviewers chQR and cavT that the paper suffers from unclear terminology and insufficient explanation of key concepts, which makes it hard to follow.

Taking everything together, I would lean toward an accept, though the clarity and positioning of the contributions would need improvement. I strongly encourage the authors to incorporate the new experiments and discussions from the rebuttal into the revised paper.

**Reviewer Concerns:**

- Both Reviewer 77AF and UoZA gave initially positive scores and confirmed that the rebuttal resolved most of their concerns.

- Both reviewer chQR and cavT both noted that the paper would benefit from significant improvements in writing and clarity.

- Reviewer chQR stated that while most concerns were addressed, the rebuttal further confirms that the method will be limited by a quasi-periodic temporal structure.

- Reviewer cavT may still have concerns about whether FDGK truly disentangles high-frequency detail from high-frequency motion. They also questioned the necessity of some components and highlight concerns about the clarity of the writing.

**Reviewer Scores:**

- Reviewer 77AF: 6 -> 8. The initial evaluation was positive. After the rebuttal, the reviewer confirmed they would raise the score.

- Reviewer chQR: 6 -> 6. The reviewer indicated that they would keep my rating unchanged.

- Reviewer cavT: 4 -> (4 or 6). Most concerns were addressed.

- Reviewer UoZA: 6 -> 6. The initial evaluation was positive.

---

### Decision · Program_Chairs · 2026-01-26

Accept (Poster)